# FLOW MATCHING FOR GENERATIVE MODELING

**Yaron Lipman**[1,2]  **Ricky T. Q. Chen**[1]  **Heli Ben-Hamu**[2]  **Maximilian Nickel**[1]  **Matt Le**[1]
[1]Meta AI (FAIR)  [2]Weizmann Institute of Science

## ABSTRACT

We introduce a new paradigm for generative modeling built on Continuous Normalizing Flows (CNFs), allowing us to train CNFs at unprecedented scale. Specifically, we present the notion of Flow Matching (FM), a simulation-free approach for training CNFs based on regressing vector fields of fixed conditional probability paths. Flow Matching is compatible with a general family of Gaussian probability paths for transforming between noise and data samples—which subsumes existing diffusion paths as specific instances. Interestingly, we find that employing FM with diffusion paths results in a more robust and stable alternative for training diffusion models. Furthermore, Flow Matching opens the door to training CNFs with other, non-diffusion probability paths. An instance of particular interest is using Optimal Transport (OT) displacement interpolation to define the conditional probability paths. These paths are more efficient than diffusion paths, provide faster training and sampling, and result in better generalization. Training CNFs using Flow Matching on ImageNet leads to consistently better performance than alternative diffusion-based methods in terms of both likelihood and sample quality, and allows fast and reliable sample generation using off-the-shelf numerical ODE solvers.

## 1 INTRODUCTION

Deep generative models are a class of deep learning algorithms aimed at estimating and sampling from an unknown data distribution. The recent influx of amazing advances in generative modeling, *e.g.*, for image generation Ramesh et al. (2022); Rombach et al. (2022), is mostly facilitated by the scalable and relatively stable training of diffusion-based models Ho et al. (2020); Song et al. (2020b). However, the restriction to simple diffusion processes leads to a rather confined space of sampling probability paths, resulting in very long training times and the need to adopt specialized methods (*e.g.*, Song et al. (2020a); Zhang & Chen (2022)) for efficient sampling.

In this work we consider the general and deterministic framework of Continuous Normalizing Flows (CNFs; Chen et al. (2018)). CNFs are capable of modeling arbitrary probability path and are in particular known to encompass the probability paths modeled by diffusion processes (Song et al., 2021).

However, aside from diffusion that can be trained efficiently via, *e.g.*, denoising score matching (Vincent, 2011), no scalable CNF training algorithms are known. Indeed, maximum likelihood training (*e.g.*, Grathwohl et al. (2018)) require expensive numerical ODE simulations, while existing simulation-free methods either involve intractable integrals (Rozen et al., 2021) or biased gradients (Ben-Hamu et al., 2022).

The goal of this work is to propose Flow Matching (FM), an efficient simulation-free approach to training CNF models, allowing the adoption of general probability paths to supervise CNF training. Importantly, FM breaks the barriers for scalable CNF training beyond diffusion, and sidesteps the need to reason about diffusion processes to directly work with probability paths.

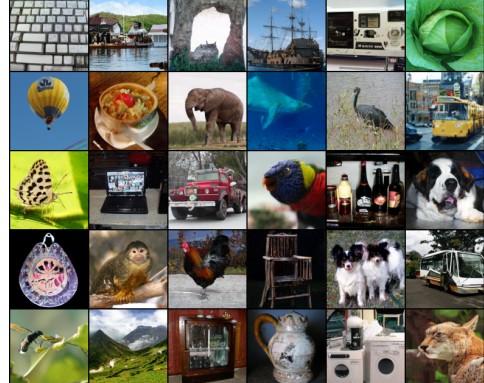

Figure 1: Unconditional ImageNet-128 samples of a CNF trained using Flow Matching with Optimal Transport probability paths.

In particular, we propose the Flow Matching objective (Section 3), a simple and intuitive training objective to regress onto a target vector field that generates a desired probability path. We first show that we can construct such target vector fields through per-example (*i.e.*, conditional) formulations. Then, inspired by denoising score matching, we show that a per-example training objective, termed Conditional Flow Matching (CFM), provides equivalent gradients and does not require explicit knowledge of the intractable target vector field. Furthermore, we discuss a general family of per-example probability paths (Section 4) that can be used for Flow Matching, which subsumes existing diffusion paths as special instances. Even on diffusion paths, we find that using FM provides more robust and stable training, and achieves superior performance compared to score matching. Furthermore, this family of probability paths also includes a particularly interesting case: the vector field that corresponds to an Optimal Transport (OT) displacement interpolant (McCann, 1997). We find that conditional OT paths are simpler than diffusion paths, forming straight line trajectories whereas diffusion paths result in curved paths. These properties seem to empirically translate to faster training, faster generation, and better performance.

We empirically validate Flow Matching and the construction via Optimal Transport paths on ImageNet, a large and highly diverse image dataset. We find that we can easily train models to achieve favorable performance in both likelihood estimation and sample quality amongst competing diffusion-based methods. Furthermore, we find that our models produce better trade-offs between computational cost and sample quality compared to prior methods. Figure 1 depicts selected unconditional ImageNet $128 \times 128$ samples from our model.

## 2   PRELIMINARIES: CONTINUOUS NORMALIZING FLOWS

Let $\mathbb{R}^d$ denote the data space with data points $x = (x^1, \ldots, x^d) \in \mathbb{R}^d$. Two important objects we use in this paper are: the *probability density path* $p : [0, 1] \times \mathbb{R}^d \to \mathbb{R}_{>0}$, which is a time dependent[1] probability density function, *i.e.*, $\int p_t(x)dx = 1$, and a *time-dependent vector field*, $v : [0, 1] \times \mathbb{R}^d \to \mathbb{R}^d$. A vector field $v_t$ can be used to construct a time-dependent diffeomorphic map, called a *flow*, $\phi : [0, 1] \times \mathbb{R}^d \to \mathbb{R}^d$, defined via the ordinary differential equation (ODE):

$$\frac{d}{dt}\phi_t(x) = v_t(\phi_t(x)) \tag{1}$$

$$\phi_0(x) = x \tag{2}$$

Previously, Chen et al. (2018) suggested modeling the vector field $v_t$ with a neural network, $v_t(x; \theta)$, where $\theta \in \mathbb{R}^p$ are its learnable parameters, which in turn leads to a deep parametric model of the flow $\phi_t$, called a *Continuous Normalizing Flow* (CNF). A CNF is used to reshape a simple prior density $p_0$ (*e.g.*, pure noise) to a more complicated one, $p_1$, via the push-forward equation

$$p_t = [\phi_t]_* p_0 \tag{3}$$

where the push-forward (or change of variables) operator $*$ is defined by

$$[\phi_t]_* p_0(x) = p_0(\phi_t^{-1}(x)) \det\left[\frac{\partial \phi_t^{-1}}{\partial x}(x)\right]. \tag{4}$$

A vector field $v_t$ is said to *generate* a probability density path $p_t$ if its flow $\phi_t$ satisfies equation 3. One practical way to test if a vector field generates a probability path is using the continuity equation, which is a key component in our proofs, see Appendix A. We recap more information on CNFs, in particular how to compute the probability $p_1(x)$ at an arbitrary point $x \in \mathbb{R}^d$ in Appendix C.

## 3   FLOW MATCHING

Let $x_1$ denote a random variable distributed according to some unknown data distribution $q(x_1)$. We assume we only have access to data samples from $q(x_1)$ but have no access to the density function itself. Furthermore, we let $p_t$ be a probability path such that $p_0 = p$ is a simple distribution, *e.g.*, the standard normal distribution $p(x) = \mathcal{N}(x|0, I)$, and let $p_1$ be approximately equal in distribution to $q$. We will later discuss how to construct such a path. The Flow Matching objective is then designed to match this target probability path, which will allow us to flow from $p_0$ to $p_1$.

---

[1]We use subscript to denote the time parameter, *e.g.*, $p_t(x)$.

Given a target probability density path $p_t(x)$ and a corresponding vector field $u_t(x)$, which generates $p_t(x)$, we define the Flow Matching (FM) objective as

$$\mathcal{L}_{\text{FM}}(\theta) = \mathbb{E}_{t, p_t(x)} \| v_t(x) - u_t(x) \|^2, \tag{5}$$

where $\theta$ denotes the learnable parameters of the CNF vector field $v_t$ (as defined in Section 2), $t \sim \mathcal{U}[0, 1]$ (uniform distribution), and $x \sim p_t(x)$. Simply put, the FM loss regresses the vector field $u_t$ with a neural network $v_t$. Upon reaching zero loss, the learned CNF model will generate $p_t(x)$.

Flow Matching is a simple and attractive objective, but naïvely on its own, it is intractable to use in practice since we have no prior knowledge for what an appropriate $p_t$ and $u_t$ are. There are many choices of probability paths that can satisfy $p_1(x) \approx q(x)$, and more importantly, we generally don't have access to a closed form $u_t$ that generates the desired $p_t$. In this section, we show that we can construct both $p_t$ and $u_t$ using probability paths and vector fields that are only defined *per sample*, and an appropriate method of aggregation provides the desired $p_t$ and $u_t$. Furthermore, this construction allows us to create a much more tractable objective for Flow Matching.

## 3.1 CONSTRUCTING $p_t, u_t$ FROM CONDITIONAL PROBABILITY PATHS AND VECTOR FIELDS

A simple way to construct a target probability path is via a mixture of simpler probability paths: Given a particular data sample $x_1$ we denote by $p_t(x|x_1)$ a *conditional probability path* such that it satisfies $p_0(x|x_1) = p(x)$ at time $t = 0$, and we design $p_1(x|x_1)$ at $t = 1$ to be a distribution concentrated around $x = x_1$, e.g., $p_1(x|x_1) = \mathcal{N}(x|x_1, \sigma^2 I)$, a normal distribution with $x_1$ mean and a sufficiently small standard deviation $\sigma > 0$. Marginalizing the conditional probability paths over $q(x_1)$ give rise to *the marginal probability path*

$$p_t(x) = \int p_t(x|x_1) q(x_1) dx_1, \tag{6}$$

where in particular at time $t = 1$, the marginal probability $p_1$ is a mixture distribution that closely approximates the data distribution $q$,

$$p_1(x) = \int p_1(x|x_1) q(x_1) dx_1 \approx q(x). \tag{7}$$

Interestingly, we can also define a *marginal vector field*, by "marginalizing" over the conditional vector fields in the following sense (we assume $p_t(x) > 0$ for all $t$ and $x$):

$$u_t(x) = \int u_t(x|x_1) \frac{p_t(x|x_1) q(x_1)}{p_t(x)} dx_1, \tag{8}$$

where $u_t(\cdot|x_1) : \mathbb{R}^d \to \mathbb{R}^d$ is a conditional vector field that generates $p_t(\cdot|x_1)$. It may not seem apparent, but this way of aggregating the conditional vector fields actually results in the correct vector field for modeling the marginal probability path.

Our first key observation is this:

> *The marginal vector field (equation 8) generates the marginal probability path (equation 6).*

This provides a surprising connection between the conditional VFs (those that generate conditional probability paths) and the marginal VF (those that generate the marginal probability path). This connection allows us to break down the unknown and intractable marginal VF into simpler conditional VFs, which are much simpler to define as these only depend on a single data sample. We formalize this in the following theorem.

**Theorem 1.** *Given vector fields $u_t(x|x_1)$ that generate conditional probability paths $p_t(x|x_1)$, for any distribution $q(x_1)$, the marginal vector field $u_t$ in equation 8 generates the marginal probability path $p_t$ in equation 6, i.e., $u_t$ and $p_t$ satisfy the continuity equation (equation 25).*

The full proofs for our theorems are all provided in Appendix B. Theorem 1 can also be derived from the Diffusion Mixture Representation Theorem in Peluchetti (2021) that provides a formula for the marginal drift and diffusion coefficients in diffusion SDEs.

## 3.2 CONDITIONAL FLOW MATCHING

Unfortunately, due to the intractable integrals in the definitions of the marginal probability path and VF (equations 6 and 8), it is still intractable to compute $u_t$, and consequently, intractable to naïvely compute an unbiased estimator of the original Flow Matching objective. Instead, we propose a simpler objective, which surprisingly will result in the same optima as the original objective. Specifically, we consider the *Conditional Flow Matching* (CFM) objective,

$$\mathcal{L}_{\text{CFM}}(\theta) = \mathbb{E}_{t,q(x_1),p_t(x|x_1)} \big\| v_t(x) - u_t(x|x_1) \big\|^2, \tag{9}$$

where $t \sim \mathcal{U}[0,1]$, $x_1 \sim q(x_1)$, and now $x \sim p_t(x|x_1)$. Unlike the FM objective, the CFM objective allows us to easily sample unbiased estimates as long as we can efficiently sample from $p_t(x|x_1)$ and compute $u_t(x|x_1)$, both of which can be easily done as they are defined on a per-sample basis. Our second key observation is therefore:

*The FM (equation 5) and CFM (equation 9) objectives have identical gradients w.r.t. $\theta$.*

That is, optimizing the CFM objective is equivalent (in expectation) to optimizing the FM objective. Consequently, this allows us to train a CNF to generate the marginal probability path $p_t$—which in particular, approximates the unknown data distribution $q$ at $t{=}1$— without ever needing access to either the marginal probability path or the marginal vector field. We simply need to design suitable *conditional* probability paths and vector fields. We formalize this property in the following theorem.

**Theorem 2.** *Assuming that $p_t(x) > 0$ for all $x \in \mathbb{R}^d$ and $t \in [0,1]$, then, up to a constant independent of $\theta$, $\mathcal{L}_{CFM}$ and $\mathcal{L}_{FM}$ are equal. Hence, $\nabla_\theta \mathcal{L}_{FM}(\theta) = \nabla_\theta \mathcal{L}_{CFM}(\theta)$.*

## 4 CONDITIONAL PROBABILITY PATHS AND VECTOR FIELDS

The Conditional Flow Matching objective works with any choice of conditional probability path and conditional vector fields. In this section, we discuss the construction of $p_t(x|x_1)$ and $u_t(x|x_1)$ for a general family of Gaussian conditional probability paths. Namely, we consider conditional probability paths of the form

$$p_t(x|x_1) = \mathcal{N}(x \mid \mu_t(x_1), \sigma_t(x_1)^2 I), \tag{10}$$

where $\mu : [0,1] \times \mathbb{R}^d \to \mathbb{R}^d$ is the time-dependent mean of the Gaussian distribution, while $\sigma : [0,1] \times \mathbb{R} \to \mathbb{R}_{>0}$ describes a time-dependent scalar standard deviation (std). We set $\mu_0(x_1) = 0$ and $\sigma_0(x_1) = 1$, so that all conditional probability paths converge to the same standard Gaussian noise distribution at $t = 0$, $p(x) = \mathcal{N}(x|0, I)$. We then set $\mu_1(x_1) = x_1$ and $\sigma_1(x_1) = \sigma_{\min}$, which is set sufficiently small so that $p_1(x|x_1)$ is a concentrated Gaussian distribution centered at $x_1$.

There is an infinite number of vector fields that generate any particular probability path (*e.g.*, by adding a divergence free component to the continuity equation, see equation 25), but the vast majority of these is due to the presence of components that leave the underlying distribution invariant—for instance, rotational components when the distribution is rotation-invariant—leading to unnecessary extra compute. We decide to use the simplest vector field corresponding to a canonical transformation for Gaussian distributions. Specifically, consider the flow (conditioned on $x_1$)

$$\psi_t(x) = \sigma_t(x_1)x + \mu_t(x_1). \tag{11}$$

When $x$ is distributed as a standard Gaussian, $\psi_t(x)$ is the affine transformation that maps to a normally-distributed random variable with mean $\mu_t(x_1)$ and std $\sigma_t(x_1)$. That is to say, according to equation 4, $\psi_t$ pushes the noise distribution $p_0(x|x_1) = p(x)$ to $p_t(x|x_1)$, *i.e.*,

$$[\psi_t]_* \, p(x) = p_t(x|x_1). \tag{12}$$

This flow then provides a vector field that generates the conditional probability path:

$$\frac{d}{dt}\psi_t(x) = u_t(\psi_t(x)|x_1). \tag{13}$$

Reparameterizing $p_t(x|x_1)$ in terms of just $x_0$ and plugging equation 13 in the CFM loss we get

$$\mathcal{L}_{\text{CFM}}(\theta) = \mathbb{E}_{t,q(x_1),p(x_0)} \Big\| v_t(\psi_t(x_0)) - \frac{d}{dt}\psi_t(x_0) \Big\|^2. \tag{14}$$

Since $\psi_t$ is a simple (invertible) affine map we can use equation 13 to solve for $u_t$ in a closed form. Let $f'$ denote the derivative with respect to time, *i.e.*, $f' = \frac{d}{dt}f$, for a time-dependent function $f$.

**Theorem 3.** *Let $p_t(x|x_1)$ be a Gaussian probability path as in equation 10, and $\psi_t$ its corresponding flow map as in equation 11. Then, the unique vector field that defines $\psi_t$ has the form:*

$$u_t(x|x_1) = \frac{\sigma'_t(x_1)}{\sigma_t(x_1)}(x - \mu_t(x_1)) + \mu'_t(x_1). \tag{15}$$

*Consequently, $u_t(x|x_1)$ generates the Gaussian path $p_t(x|x_1)$.*

### 4.1 SPECIAL INSTANCES OF GAUSSIAN CONDITIONAL PROBABILITY PATHS

Our formulation is fully general for arbitrary functions $\mu_t(x_1)$ and $\sigma_t(x_1)$, and we can set them to any differentiable function satisfying the desired boundary conditions. We first discuss the special cases that recover probability paths corresponding to previously-used diffusion processes. Since we directly work with probability paths, we can simply depart from reasoning about diffusion processes altogether. Therefore, in the second example below, we directly formulate a probability path based on the Wasserstein-2 optimal transport solution as an interesting instance.

**Example I: Diffusion conditional VFs.** Diffusion models start with data points and gradually add noise until it approximates pure noise. These can be formulated as stochastic processes, which have strict requirements in order to obtain closed form representation at arbitrary times $t$, resulting in Gaussian conditional probability paths $p_t(x|x_1)$ with specific choices of mean $\mu_t(x_1)$ and std $\sigma_t(x_1)$ (Sohl-Dickstein et al., 2015; Ho et al., 2020; Song et al., 2020b). For example, the reversed (noise→data) Variance Exploding (VE) path has the form

$$p_t(x|x_1) = \mathcal{N}(x|x_1, \sigma^2_{1-t}I), \tag{16}$$

where $\sigma_t$ is an increasing function, $\sigma_0 = 0$, and $\sigma_1 \gg 1$. Next, equation 16 provides the choices of $\mu_t(x_1) = x_1$ and $\sigma_t(x_1) = \sigma_{1-t}$. Plugging these into equation 15 of Theorem 3 we get

$$u_t(x|x_1) = -\frac{\sigma'_{1-t}}{\sigma_{1-t}}(x - x_1). \tag{17}$$

The reversed (noise→data) Variance Preserving (VP) diffusion path has the form

$$p_t(x|x_1) = \mathcal{N}\left(x \mid \alpha_{1-t}x_1, \left(1 - \alpha^2_{1-t}\right)I\right), \text{where } \alpha_t = e^{-\frac{1}{2}T(t)}, T(t) = \int_0^t \beta(s)ds, \tag{18}$$

and $\beta$ is the noise scale function. Equation 18 provides the choices of $\mu_t(x_1) = \alpha_{1-t}x_1$ and $\sigma_t(x_1) = \sqrt{1 - \alpha^2_{1-t}}$. Plugging these into equation 15 of Theorem 3 we get

$$u_t(x|x_1) = \frac{\alpha'_{1-t}}{1 - \alpha^2_{1-t}}(\alpha_{1-t}x - x_1) = -\frac{T'(1-t)}{2}\left[\frac{e^{-T(1-t)}x - e^{-\frac{1}{2}T(1-t)}x_1}{1 - e^{-T(1-t)}}\right]. \tag{19}$$

Our construction of the conditional VF $u_t(x|x_1)$ does in fact coincide with the vector field previously used in the deterministic probability flow (Song et al. (2020b), equation 13) when restricted to these conditional diffusion processes; see details in Appendix D. Nevertheless, combining the diffusion conditional VF with the Flow Matching objective offers an attractive training alternative—which we find to be more stable and robust in our experiments—to existing score matching approaches.

Another important observation is that, as these probability paths were previously derived as solutions of diffusion processes, they do not actually reach a true noise distribution in finite time. In practice, $p_0(x)$ is simply approximated by a suitable Gaussian distribution for sampling and likelihood evaluation. Instead, our construction provides full control over the probability path, and we can just directly set $\mu_t$ and $\sigma_t$, as we will do next.

**Example II: Optimal Transport conditional VFs.** An arguably more natural choice for conditional probability paths is to define the mean and the std to simply change linearly in time, *i.e.*,

$$\mu_t(x) = tx_1, \text{ and } \sigma_t(x) = 1 - (1 - \sigma_{\min})t. \tag{20}$$

According to Theorem 3 this path is generated by the VF

$$u_t(x|x_1) = \frac{x_1 - (1 - \sigma_{\min})x}{1 - (1 - \sigma_{\min})t}, \tag{21}$$

$t = 0.0 \qquad t = 1/3 \qquad t = 2/3 \qquad t = 1.0 \qquad\qquad t = 0.0 \qquad t = 1/3 \qquad t = 2/3 \qquad t = 1.0$

Diffusion path – conditional score function $\qquad$ OT path – conditional vector field

Figure 2: Compared to the diffusion path's conditional score function, the OT path's conditional vector field has constant direction in time and is arguably simpler to fit with a parametric model. Note the blue color denotes larger magnitude while red color denotes smaller magnitude.

which, in contrast to the diffusion conditional VF (equation 19), is defined for all $t \in [0, 1]$. The conditional flow that corresponds to $u_t(x|x_1)$ is

$$\psi_t(x) = (1 - (1 - \sigma_{\min})t)x + tx_1, \tag{22}$$

and in this case, the CFM loss (see equations 9, 14) takes the form:

$$\mathcal{L}_{\mathrm{CFM}}(\theta) = \mathbb{E}_{t,q(x_1),p(x_0)}\left\| v_t(\psi_t(x_0)) - \left(x_1 - (1 - \sigma_{\min})x_0\right)\right\|^2. \tag{23}$$

Allowing the mean and std to change linearly not only leads to simple and intuitive paths, but it is actually also optimal in the following sense. The conditional flow $\psi_t(x)$ is in fact the Optimal Transport (OT) *displacement map* between the two Gaussians $p_0(x|x_1)$ and $p_1(x|x_1)$. The OT *interpolant*, which is a probability path, is defined to be (see Definition 1.1 in McCann (1997)):

$$p_t = [(1 - t)\mathrm{id} + t\psi]_\star p_0 \tag{24}$$

where $\psi : \mathbb{R}^d \to \mathbb{R}^d$ is the OT map pushing $p_0$ to $p_1$, id denotes the identity map, *i.e.*, $\mathrm{id}(x) = x$, and $(1 - t)\mathrm{id} + t\psi$ is called the OT displacement map. Example 1.7 in McCann (1997) shows, that in our case of two Gaussians where the first is a standard one, the OT displacement map takes the form of equation 22.

Intuitively, particles under the OT displacement map always move in straight line trajectories and with constant speed. Figure 3 depicts sampling paths for the diffusion and OT conditional VFs. Interestingly, we find that sampling trajectory from diffusion paths can "overshoot" the final sample, resulting in unnecessary backtracking, whilst the OT paths are guaranteed to stay straight.

Diffusion $\qquad$ OT

Figure 3: Diffusion and OT conditional trajectories.

Figure 2 compares the diffusion conditional score function (the regression target in a typical diffusion methods), *i.e.*, $\nabla \log p_t(x|x_1)$ with $p_t$ defined as in equation 18, with the OT conditional VF (equation 21). The start ($p_0$) and end ($p_1$) Gaussians are identical in both examples. An interesting observation is that the OT VF has a constant direction in time, which arguably leads to a simpler regression task. This property can also be verified directly from equation 21 as the VF can be written in the form $u_t(x|x_1) = g(t)h(x|x_1)$. Figure 8 in the Appendix shows a visualization of the Diffusion VF. Lastly, we note that although the conditional flow is optimal, this by no means imply that the marginal VF is an optimal transport solution. Nevertheless, we expect the marginal vector field to remain relatively simple.

## 5 RELATED WORK

Continuous Normalizing Flows were introduced in (Chen et al., 2018) as a continuous-time version of Normalizing Flows (see *e.g.*, Kobyzev et al. (2020); Papamakarios et al. (2021) for an overview). Originally, CNFs are trained with the maximum likelihood objective, but this involves expensive ODE simulations for the forward and backward propagation, resulting in high time complexity due to the sequential nature of ODE simulations. Although some works demonstrated the capability of CNF generative models for image synthesis (Grathwohl et al., 2018), scaling up to very high dimensional images is inherently difficult. A number of works attempted to regularize the ODE to be easier to solve, *e.g.*, using augmentation (Dupont et al., 2019), adding regularization terms (Yang & Karniadakis, 2019; Finlay et al., 2020; Onken et al., 2021; Tong et al., 2020; Kelly et al., 2020), or stochastically sampling the integration interval (Du et al., 2022). These works merely aim to regularize the ODE but do not change the fundamental training algorithm.

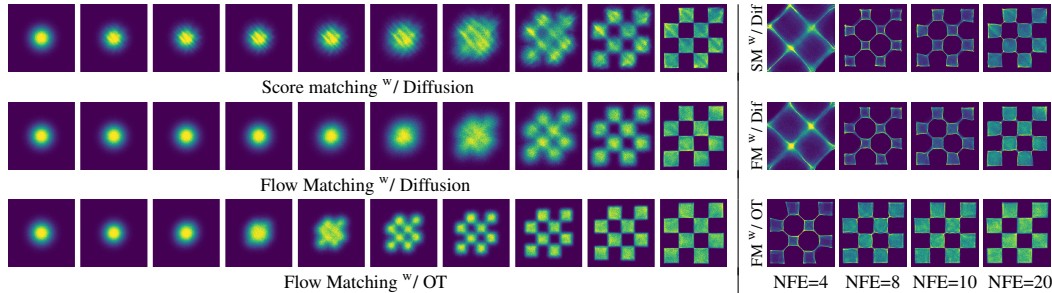

Figure 4: (*left*) Trajectories of CNFs trained with different objectives on 2D checkerboard data. The OT path introduces the checkerboard pattern much earlier, while FM results in more stable training. (*right*) FM with OT results in more efficient sampling, solved using the midpoint scheme.

In order to speed up CNF training, some works have developed simulation-free CNF training frameworks by explicitly designing the target probability path and the dynamics. For instance, Rozen et al. (2021) consider a linear interpolation between the prior and the target density but involves integrals that were difficult to estimate in high dimensions, while Ben-Hamu et al. (2022) consider general probability paths similar to this work but suffers from biased gradients in the stochastic minibatch regime. In contrast, the Flow Matching framework allows simulation-free training with unbiased gradients and readily scales to very high dimensions.

Another approach to simulation-free training relies on the construction of a diffusion process to indirectly define the target probability path (Sohl-Dickstein et al., 2015; Ho et al., 2020; Song & Ermon, 2019). Song et al. (2020b) shows that diffusion models are trained using denoising score matching (Vincent, 2011), a conditional objective that provides unbiased gradients with respect to the score matching objective. Conditional Flow Matching draws inspiration from this result, but generalizes to matching vector fields directly. Due to the ease of scalability, diffusion models have received increased attention, producing a variety of improvements such as loss-rescaling (Song et al., 2021), adding classifier guidance along with architectural improvements (Dhariwal & Nichol, 2021), and learning the noise schedule (Nichol & Dhariwal, 2021; Kingma et al., 2021). However, (Nichol & Dhariwal, 2021) and (Kingma et al., 2021) only consider a restricted setting of Gaussian conditional paths defined by simple diffusion processes with a single parameter—in particular, it does not include our conditional OT path. In an another line of works, (De Bortoli et al., 2021; Wang et al., 2021; Peluchetti, 2021) proposed finite time diffusion constructions via diffusion bridges theory resolving the approximation error incurred by infinite time denoising constructions. While existing works make use of a connection between diffusion processes and continuous normalizing flows with the same probability path (Maoutsa et al., 2020b; Song et al., 2020b; 2021), our work allows us to generalize beyond the class of probability paths modeled by simple diffusion. With our work, it is possible to completely sidestep the diffusion process construction and reason directly with probability paths, while still retaining efficient training and log-likelihood evaluations. Lastly, concurrently to our work (Liu et al., 2022; Albergo & Vanden-Eijnden, 2022) arrived at similar conditional objectives for simulation-free training of CNFs, while Neklyudov et al. (2023) derived an implicit objective when $u_t$ is assumed to be a gradient field.

# 6 EXPERIMENTS

We explore the empirical benefits of using Flow Matching on the image datasets of CIFAR-10 (Krizhevsky et al., 2009) and ImageNet at resolutions 32, 64, and 128 (Chrabaszcz et al., 2017; Deng et al., 2009). We also ablate the choice of diffusion path in Flow Matching, particularly between the standard variance preserving diffusion path and the optimal transport path. We discuss how sample generation is improved by directly parameterizing the generating vector field and using the Flow Matching objective. Lastly we show Flow Matching can also be used in the conditional generation setting. Unless otherwise specified, we evaluate likelihood and samples from the model using dopri5 (Dormand & Prince, 1980) at absolute and relative tolerances of 1e-5. Generated samples can be found in the Appendix, and all implementation details are in Appendix E.

| Model | CIFAR-10 | | | ImageNet 32×32 | | | ImageNet 64×64 | | | | Model | ImageNet 128×128 | |
|---|---|---|---|---|---|---|---|---|---|---|---|---|---|
| | NLL↓ | FID↓ | NFE↓ | NLL↓ | FID↓ | NFE↓ | NLL↓ | FID↓ | NFE↓ | | | NLL↓ | FID↓ |
| *Ablations* | | | | | | | | | | | MGAN (Hoang et al., 2018) | – | 58.9 |
| DDPM | 3.12 | 7.48 | 274 | 3.54 | 6.99 | 262 | 3.32 | 17.36 | 264 | | PacGAN2 (Lin et al., 2018) | – | 57.5 |
| Score Matching | 3.16 | 19.94 | 242 | 3.56 | 5.68 | 178 | 3.40 | 19.74 | 441 | | Logo-GAN-AE (Sage et al., 2018) | – | 50.9 |
| ScoreFlow | 3.09 | 20.78 | 428 | 3.55 | 14.14 | 195 | 3.36 | 24.95 | 601 | | Self-cond. GAN (Lučić et al., 2019) | – | 41.7 |
| *Ours* | | | | | | | | | | | Uncond. BigGAN (Lučić et al., 2019) | – | 25.3 |
| FM ᵂ/ Diffusion | 3.10 | 8.06 | 183 | 3.54 | 6.37 | 193 | 3.33 | 16.88 | 187 | | PGMGAN (Armandpour et al., 2021) | – | 21.7 |
| FM ᵂ/ OT | **2.99** | **6.35** | **142** | **3.53** | **5.02** | **122** | **3.31** | **14.45** | **138** | | FM ᵂ/ OT | **2.90** | **20.9** |

Table 1: Likelihood (BPD), quality of generated samples (FID), and evaluation time (NFE) for the same model trained with different methods.

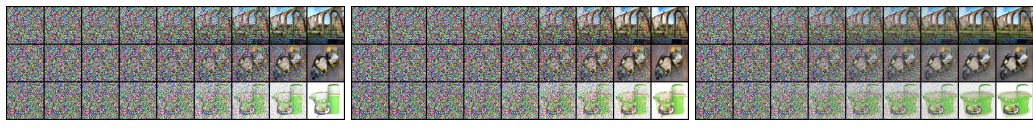

Score Matching w/ Diffusion  ·  Flow Matching w/ Diffusion  ·  Flow Matching w/ OT

Figure 6: Sample paths from the same initial noise with models trained on ImageNet 64×64. The OT path reduces noise roughly linearly, while diffusion paths visibly remove noise only towards the end of the path. Note also the differences between the generated images.

## 6.1 DENSITY MODELING AND SAMPLE QUALITY ON IMAGENET

We start by comparing the same model architecture, *i.e.*, the U-Net architecture from Dhariwal & Nichol (2021) with minimal changes, trained on CIFAR-10, and ImageNet 32/64 with different popular diffusion-based losses: DDPM from (Ho et al., 2020), Score Matching (SM) (Song et al., 2020b), and Score Flow (SF) (Song et al., 2021); see Appendix E.1 for exact details. Table 1 (left) summarizes our results alongside these baselines reporting negative log-likelihood (NLL) in units of bits per dimension (BPD), sample quality as measured by the Frechet Inception Distance (FID; Heusel et al. (2017)), and averaged number of function evaluations (NFE) required for the adaptive solver to reach its a prespecified numerical tolerance, averaged over 50k samples. All models are trained using the same architecture, hyperparameter values and number of training iterations, where baselines are allowed more iterations for better convergence. Note that these are *unconditional* models. On both CIFAR-10 and ImageNet, FM-OT consistently obtains best results across all our quantitative measures compared to competing methods. We are noticing a higher that usual FID performance in CIFAR-10 compared to previous works (Ho et al., 2020; Song et al., 2020b; 2021) that can possibly be explained by the fact that our used architecture was not optimized for CIFAR-10.

Secondly, Table 1 (right) compares a model trained using Flow Matching with the OT path on ImageNet at resolution 128×128. Our FID is state-of-the-art with the exception of IC-GAN (Casanova et al., 2021) which uses conditioning with a self-supervised ResNet50 model, and therefore is left out of this table. Figures 11, 12, 13 in the Appendix show non-curated samples from these models.

**Faster training.** While existing works train diffusion models with a very high number of iterations (*e.g.*, 1.3m and 10m iterations are reported by Score Flow and VDM, respectively), we find that Flow Matching generally converges much faster. Figure 5 shows FID curves during training of Flow Matching and all baselines for ImageNet 64×64; FM-OT is able to lower the FID faster and to a greater extent than the alternatives. For ImageNet-128 Dhariwal & Nichol (2021) train for 4.36m iterations with batch size 256, while FM (with 25% larger model) used 500k iterations with batch size 1.5k, *i.e.*, 33% less image throughput; see Table 3 for exact details. Furthermore, the cost of sampling from a model can drastically change during training for score matching, whereas the sampling cost stays constant when training with Flow Matching (Figure 10 in Appendix).

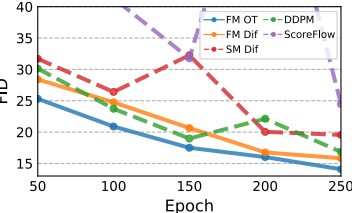

Figure 5: Image quality during training, ImageNet 64×64.

## 6.2 SAMPLING EFFICIENCY

For sampling, we first draw a random noise sample $x_0 \sim \mathcal{N}(0, I)$ then compute $\phi_1(x_0)$ by solving equation 1 with the trained VF, $v_t$, on the interval $t \in [0, 1]$ using an ODE solver. While diffusion

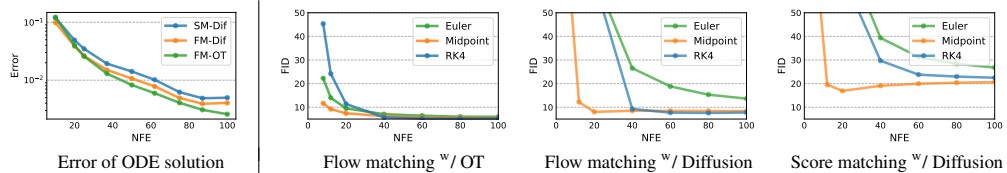

Figure 7: Flow Matching, especially when using OT paths, allows us to use fewer evaluations for sampling while retaining similar numerical error (left) and sample quality (right). Results are shown for models trained on ImageNet $32{\times}32$, and numerical errors are for the midpoint scheme.

models can also be sampled through an SDE formulation, this can be highly inefficient and many methods that propose fast samplers (*e.g.*, Song et al. (2020a); Zhang & Chen (2022)) directly make use of the ODE perspective (see Appendix D). In part, this is due to ODE solvers being much more efficient—yielding lower error at similar computational costs (Kloeden et al., 2012)—and the multitude of available ODE solver schemes. When compared to our ablation models, we find that models trained using Flow Matching with the OT path always result in the most efficient sampler, regardless of ODE solver, as demonstrated next.

**Sample paths.** We first qualitatively visualize the difference in sampling paths between diffusion and OT. Figure 6 shows samples from ImageNet-64 models using identical random seeds, where we find that the OT path model starts generating images sooner than the diffusion path models, where noise dominates the image until the very last time point. We additionally depict the probability density paths in 2D generation of a checkerboard pattern, Figure 4 (left), noticing a similar trend.

**Low-cost samples.** We next switch to fixed-step solvers and compare low ($\leq$100) NFE samples computed with the ImageNet-32 models from Table 1. In Figure 7 (left), we compare the per-pixel MSE of low NFE solutions compared with 1000 NFE solutions (we use 256 random noise seeds), and notice that the FM with OT model produces the best numerical error, in terms of computational cost, requiring roughly only 60% of the NFEs to reach the same error threshold as diffusion models. Secondly, Figure 7 (right) shows how FID changes as a result of the computational cost, where we find FM with OT is able to achieve decent FID even at very low NFE values, producing better trade-off between sample quality and cost compared to ablated models. Figure 4 (right) shows low-cost sampling effects for the 2D checkerboard experiment.

### 6.3 CONDITIONAL SAMPLING FROM LOW-RESOLUTION IMAGES

Lastly, we experimented with Flow Matching for conditional image generation. In particular, upsampling images from $64{\times}64$ to $256{\times}256$. We follow the evaluation procedure in (Saharia et al., 2022) and compute the FID of the upsampled validation images; baselines include reference (FID of original validation set), and regression. Results are in Table 2. Upsampled image samples are shown in Figures 14, 15 in the Appendix. FM-OT achieves similar PSNR and SSIM values to (Saharia et al., 2022) while

| Model | FID↓ | IS↑ | PSNR↑ | SSIM↑ |
|---|---|---|---|---|
| Reference | 1.9 | 240.8 | – | – |
| Regression | 15.2 | 121.1 | **27.9** | **0.801** |
| SR3 (Saharia et al., 2022) | 5.2 | 180.1 | 26.4 | 0.762 |
| FM ʷ/ OT | **3.4** | **200.8** | 24.7 | 0.747 |

Table 2: Image super-resolution on the ImageNet validation set.

considerably improving on FID and IS, which as argued by (Saharia et al., 2022) is a better indication of generation quality.

## 7 CONCLUSION

We introduced Flow Matching, a new simulation-free framework for training Continuous Normalizing Flow models, relying on conditional constructions to effortlessly scale to very high dimensions. Furthermore, the FM framework provides an alternative view on diffusion models, and suggests forsaking the stochastic/diffusion construction in favor of more directly specifying the probability path, allowing us to, *e.g.*, construct paths that allow faster sampling and/or improve generation. We experimentally showed the ease of training and sampling when using the Flow Matching framework, and in the future, we expect FM to open the door to allowing a multitude of probability paths (*e.g.*, non-isotropic Gaussians or more general kernels altogether).

SOCIAL RESPONSIBILITY

Along side its many positive applications, image generation can also be used for harmful proposes. Using content-controlled training sets and image validation/classification can help reduce these uses. Furthermore, the energy demand for training large deep learning models is increasing at a rapid pace (Amodei et al., 2018; Thompson et al., 2020), focusing on methods that are able to train using less gradient updates / image throughput can lead to significant time and energy savings.

ACKNOWLEDGEMENTS

We thank Qinqing Zheng for her feedback. HB is supported by a grant from Israel CHE Program for Data Science Research Centers. Additionally, we acknowledge the Python community (Van Rossum & Drake Jr, 1995; Oliphant, 2007) for developing the core set of tools that enabled this work, including PyTorch (Paszke et al., 2019), PyTorch Lightning (Falcon & team, 2019), Hydra (Yadan, 2019), Jupyter (Kluyver et al., 2016), Matplotlib (Hunter, 2007), seaborn (Waskom et al., 2018), numpy (Oliphant, 2006; Van Der Walt et al., 2011), SciPy (Jones et al., 2014), and torchdiffeq (Chen, 2018).

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

# A  THE CONTINUITY EQUATION

One method of testing if a vector field $v_t$ generates a probability path $p_t$ is the continuity equation (Villani, 2009). It is a Partial Differential Equation (PDE) providing a necessary and sufficient condition to ensuring that a vector field $v_t$ generates $p_t$,

$$\frac{d}{dt}p_t(x) + \mathrm{div}(p_t(x)v_t(x)) = 0, \tag{25}$$

where the divergence operator, $\mathrm{div}$, is defined with respect to the spatial variable $x = (x^1, \ldots, x^d)$, i.e., $\mathrm{div} = \sum_{i=1}^{d} \frac{\partial}{\partial x^i}$.

# B  THEOREM PROOFS

**Theorem 1.** *Given vector fields $u_t(x|x_1)$ that generate conditional probability paths $p_t(x|x_1)$, for any distribution $q(x_1)$, the marginal vector field $u_t$ in equation 8 generates the marginal probability path $p_t$ in equation 6, i.e., $u_t$ and $p_t$ satisfy the continuity equation (equation 25).*

*Proof.* To verify this, we check that $p_t$ and $u_t$ satisfy the continuity equation (equation 25):

$$\frac{d}{dt}p_t(x) = \int \left(\frac{d}{dt}p_t(x|x_1)\right)q(x_1)dx_1 = -\int \mathrm{div}\Big(u_t(x|x_1)p_t(x|x_1)\Big)q(x_1)dx_1$$

$$= -\mathrm{div}\Big(\int u_t(x|x_1)p_t(x|x_1)q(x_1)dx_1\Big) = -\mathrm{div}\Big(u_t(x)p_t(x)\Big),$$

where in the second equality we used the fact that $u_t(\cdot|x_1)$ generates $p_t(\cdot|x_1)$, in the last equality we used equation 8. Furthermore, the first and third equalities are justified by assuming the integrands satisfy the regularity conditions of the Leibniz Rule (for exchanging integration and differentiation). $\square$

**Theorem 2.** *Assuming that $p_t(x) > 0$ for all $x \in \mathbb{R}^d$ and $t \in [0,1]$, then, up to a constant independent of $\theta$, $\mathcal{L}_{CFM}$ and $\mathcal{L}_{FM}$ are equal. Hence, $\nabla_\theta \mathcal{L}_{FM}(\theta) = \nabla_\theta \mathcal{L}_{CFM}(\theta)$.*

*Proof.* To ensure existence of all integrals and to allow the changing of integration order (by Fubini's Theorem) in the following we assume that $q(x)$ and $p_t(x|x_1)$ are decreasing to zero at a sufficient speed as $\|x\| \to \infty$, and that $u_t, v_t, \nabla_\theta v_t$ are bounded.

First, using the standard bilinearity of the 2-norm we have that

$$\|v_t(x) - u_t(x)\|^2 = \|v_t(x)\|^2 - 2\langle v_t(x), u_t(x)\rangle + \|u_t(x)\|^2$$

$$\|v_t(x) - u_t(x|x_1)\|^2 = \|v_t(x)\|^2 - 2\langle v_t(x), u_t(x|x_1)\rangle + \|u_t(x|x_1)\|^2$$

Next, remember that $u_t$ is independent of $\theta$ and note that

$$\mathbb{E}_{p_t(x)}\|v_t(x)\|^2 = \int \|v_t(x)\|^2 p_t(x)dx = \int \|v_t(x)\|^2 p_t(x|x_1)q(x_1)dx_1 dx$$

$$= \mathbb{E}_{q(x_1),p_t(x|x_1)}\|v_t(x)\|^2,$$

where in the second equality we use equation 6, and in the third equality we change the order of integration. Next,

$$\mathbb{E}_{p_t(x)}\langle v_t(x), u_t(x)\rangle = \int \left\langle v_t(x), \frac{\int u_t(x|x_1)p_t(x|x_1)q(x_1)dx_1}{p_t(x)}\right\rangle p_t(x)dx$$

$$= \int \left\langle v_t(x), \int u_t(x|x_1)p_t(x|x_1)q(x_1)dx_1\right\rangle dx$$

$$= \int \langle v_t(x), u_t(x|x_1)\rangle \, p_t(x|x_1)q(x_1)dx_1 dx$$

$$= \mathbb{E}_{q(x_1),p_t(x|x_1)}\langle v_t(x), u_t(x|x_1)\rangle,$$

where in the last equality we change again the order of integration. $\square$

**Theorem 3.** *Let $p_t(x|x_1)$ be a Gaussian probability path as in equation 10, and $\psi_t$ its corresponding flow map as in equation 11. Then, the unique vector field that defines $\psi_t$ has the form:*

$$u_t(x|x_1) = \frac{\sigma_t'(x_1)}{\sigma_t(x_1)}(x - \mu_t(x_1)) + \mu_t'(x_1). \tag{15}$$

*Consequently, $u_t(x|x_1)$ generates the Gaussian path $p_t(x|x_1)$.*

*Proof.* For notational simplicity let $w_t(x) = u_t(x|x_1)$. Now consider equation 1:

$$\frac{d}{dt}\psi_t(x) = w_t(\psi_t(x)).$$

Since $\psi_t$ is invertible (as $\sigma_t(x_1) > 0$) we let $x = \psi^{-1}(y)$ and get

$$\psi_t'(\psi^{-1}(y)) = w_t(y), \tag{26}$$

where we used the apostrophe notation for the derivative to emphasis that $\psi_t'$ is evaluated at $\psi^{-1}(y)$. Now, inverting $\psi_t(x)$ provides

$$\psi_t^{-1}(y) = \frac{y - \mu_t(x_1)}{\sigma_t(x_1)}.$$

Differentiating $\psi_t$ with respect to $t$ gives

$$\psi_t'(x) = \sigma_t'(x_1)x + \mu_t'(x_1).$$

Plugging these last two equations in equation 26 we get

$$w_t(y) = \frac{\sigma_t'(x_1)}{\sigma_t(x_1)}(y - \mu_t(x_1)) + \mu_t'(x_1)$$

as required. $\qquad\square$

## C  COMPUTING PROBABILITIES OF THE CNF MODEL

We are given an arbitrary data point $x_1 \in \mathbb{R}^d$ and need to compute the model probability at that point, *i.e.*, $p_1(x_1)$. Below we recap how this can be done covering the basic relevant ODEs, the scaling of the divergence computation, taking into account data transformations (*e.g.*, centering of data), and Bits-Per-Dimension computation.

**ODE for computing $p_1(x_1)$.** The continuity equation with equation 1 lead to the instantaneous change of variable (Chen et al., 2018; Ben-Hamu et al., 2022):

$$\frac{d}{dt}\log p_t(\phi_t(x)) + \mathrm{div}(v_t(\phi_t(x))) = 0.$$

Integrating $t \in [0, 1]$ gives:

$$\log p_1(\phi_1(x)) - \log p_0(\phi_0(x)) = -\int_0^1 \mathrm{div}(v_t(\phi_t(x)))dt \tag{27}$$

Therefore, the log probability can be computed together with the flow trajectory by solving the ODE:

$$\frac{d}{dt}\begin{bmatrix}\phi_t(x) \\ f(t)\end{bmatrix} = \begin{bmatrix}v_t(\phi_t(x)) \\ -\mathrm{div}(v_t(\phi_t(x)))\end{bmatrix} \tag{28}$$

Given initial conditions

$$\begin{bmatrix}\phi_0(x) \\ f(0)\end{bmatrix} = \begin{bmatrix}x_0 \\ c\end{bmatrix}. \tag{29}$$

the solution $[\phi_t(x), f(t)]^T$ is uniquely defined (up to some mild conditions on the VF $v_t$). Denote $x_1 = \phi_1(x)$, and according to equation 27,

$$f(1) = c + \log p_1(x_1) - \log p_0(x_0). \tag{30}$$

Now, we are given an arbitrary $x_1$ and want to compute $p_1(x_1)$. For this end, we will need to solve equation 28 in reverse. That is,

$$\frac{d}{ds}\begin{bmatrix}\phi_{1-s}(x)\\ f(1-s)\end{bmatrix} = \begin{bmatrix}-v_{1-s}(\phi_{1-s}(x))\\ \mathrm{div}(v_{1-s}(\phi_{1-s}(x)))\end{bmatrix} \tag{31}$$

and we solve this equation for $s \in [0,1]$ with the initial conditions at $s = 0$:

$$\begin{bmatrix}\phi_1(x)\\ f(1)\end{bmatrix} = \begin{bmatrix}x_1\\ 0\end{bmatrix}. \tag{32}$$

From uniqueness of ODEs, the solution will be identical to the solution of equation 28 with initial conditions in equation 29 where $c = \log p_0(x_0) - \log p_1(x_1)$. This can be seen from equation 30 and setting $f(1) = 0$. Therefore we get that

$$f(0) = \log p_0(x_0) - \log p_1(x_1)$$

and consequently

$$\log p_1(x_1) = \log p_0(x_0) - f(0). \tag{33}$$

To summarize, to compute $p_1(x_1)$ we first solve the ODE in equation 31 with initial conditions in equation 32, and the compute equation 33.

**Unbiased estimator to $p_1(x_1)$.** Solving equation 31 requires computation of div of VFs in $\mathbb{R}^d$ which is costly. Grathwohl et al. (2018) suggest to replace the divergence by the (unbiased) Hutchinson trace estimator,

$$\frac{d}{ds}\begin{bmatrix}\phi_{1-s}(x)\\ \tilde{f}(1-s)\end{bmatrix} = \begin{bmatrix}-v_{1-s}(\phi_{1-s}(x))\\ z^T Dv_{1-s}(\phi_{1-s}(x))z\end{bmatrix}, \tag{34}$$

where $z \in \mathbb{R}^d$ is a sample from a random variable such that $\mathbb{E}zz^T = I$. Solving the ODE in equation 34 exactly (in practice, with a small controlled error) with initial conditions in equation 32 leads to

$$\begin{aligned}\mathbb{E}_z\left[\log p_0(x_0) - \tilde{f}(0)\right] &= \log p_0(x_0) - \mathbb{E}_z\left[\tilde{f}(0) - \tilde{f}(1)\right]\\ &= \log p_0(x_0) - \mathbb{E}_z\left[\int_0^1 z^T Dv_{1-s}(\phi_{1-s}(x))z\, ds\right]\\ &= \log p_0(x_0) - \int_0^1 \mathbb{E}_z\left[z^T Dv_{1-s}(\phi_{1-s}(x))z\right] ds\\ &= \log p_0(x_0) - \int_0^1 \mathrm{div}(v_{1-s}(\phi_{1-s}(x)))ds\\ &= \log p_0(x_0) - (f(0) - f(1))\\ &= \log p_0(x_0) - (\log p_0(x_0) - \log p_1(x_1))\\ &= \log p_1(x_1),\end{aligned}$$

where in the third equality we switched order of integration assuming the sufficient condition of Fubini's theorem hold, and in the previous to last equality we used equation 30. Therefore the random variable

$$\log p_0(x_0) - \tilde{f}(0) \tag{35}$$

is an unbiased estimator for $\log p_1(x_1)$. To summarize, for a scalable unbiased estimation of $p_1(x_1)$ we first solve the ODE in equation 34 with initial conditions in equation 32, and then output equation 35.

**Transformed data.** Often, before training our generative model we transform the data, *e.g.*, we scale and/or translate the data. Such a transformation is denoted by $\varphi^{-1} : \mathbb{R}^d \to \mathbb{R}^d$ and our generative model becomes a composition

$$\psi(x) = \varphi \circ \phi(x)$$

where $\phi : \mathbb{R}^d \to \mathbb{R}^d$ is the model we train. Given a prior probability $p_0$ we have that the push forward of this probability under $\psi$ (equation 3 and equation 4) takes the form

$$p_1(x) = \psi_* p_0(x) = p_0(\phi^{-1}(\varphi^{-1}(x))) \det\left[D\phi^{-1}(\varphi^{-1}(x))\right] \det\left[D\varphi^{-1}(x)\right]$$
$$= \left(\phi_* p_0(\varphi^{-1}(x))\right) \det\left[D\varphi^{-1}(x)\right]$$

and therefore

$$\log p_1(x) = \log \phi_* p_0(\varphi^{-1}(x)) + \log \det\left[D\varphi^{-1}(x)\right].$$

For images $d = H \times W \times 3$ we consider a transform $\phi$ that maps each pixel value from $[-1, 1]$ to $[0, 256]$. Therefore,

$$\varphi(y) = 2^7(y + 1),$$

and

$$\varphi^{-1}(x) = 2^{-7}x - 1.$$

For this case we have

$$\log p_1(x) = \log \phi_* p_0(\varphi^{-1}(x)) - 7d \log 2. \tag{36}$$

**Bits-Per-Dimension (BPD) computation.** BPD is defined by

$$\text{BPD} = \mathbb{E}_{x_1}\left[-\frac{\log_2 p_1(x_1)}{d}\right] = \mathbb{E}_{x_1}\left[-\frac{\log p_1(x_1)}{d \log 2}\right] \tag{37}$$

Following equation 36 we get

$$\text{BPD} = -\frac{\log \phi_* p_0(\varphi^{-1}(x))}{d \log 2} + 7.$$

and $\log \phi_* p_0(\varphi^{-1}(x))$ is approximated using the unbiased estimator in equation 35 over the transformed data $\varphi^{-1}(x_1)$. Averaging the unbiased estimator on a large test test $x_1$ provides a good approximation to the test set BPD.

## D    DIFFUSION CONDITIONAL VECTOR FIELDS

We derive the vector field governing the Probability Flow ODE (equation 13 in Song et al. (2020b)) for the VE and VP diffusion paths (equation 18) and note that it coincides with the conditional vector fields we derive using Theorem 3, namely the vector fields defined in equations 16 and 19.

We start with a short primer on how to find a conditional vector field for the probability path described by the Fokker-Planck equation, then instantiate it for the VE and VP probability paths.

Since in the diffusion literature the diffusion process runs from data at time $t = 0$ to noise at time $t = 1$, we will need the following lemma to translate the diffusion VFs to our convention of $t = 0$ corresponds to noise and $t = 1$ corresponds to data:

**Lemma 1.** *Consider a flow defined by a vector field $u_t(x)$ generating probability density path $p_t(x)$. Then, the vector field $\tilde{u}_t(x) = -u_{1-t}(x)$ generates the path $\tilde{p}_t(x) = p_{1-t}(x)$ when initiated from $\tilde{p}_0(x) = p_1(x)$.*

*Proof.* We use the continuity equation (equation 25):

$$\frac{d}{dt}\tilde{p}_t(x) = \frac{d}{dt}p_{1-t}(x) = -p'_{1-t}(x)$$
$$= \text{div}(p_{1-t}(x)u_{1-t}(x))$$
$$= -\text{div}(\tilde{p}_t(x)(-u_{1-t}(x)))$$

and therefore $\tilde{u}_t(x) = -u_{1-t}(x)$ generates $\tilde{p}_t(x)$. $\qquad\square$

**Conditional VFs for Fokker-Planck probability paths**  Consider a Stochastic Differential Equation (SDE) of the standard form

$$dy = f_t dt + g_t dw \tag{38}$$

with time parameter $t$, drift $f_t$, diffusion coefficient $g_t$, and $dw$ is the Wiener process. The solution $y_t$ to the SDE is a stochastic process, *i.e.*, a continuous time-dependent random variable, the probability density of which, $p_t(y_t)$, is characterized by the Fokker-Planck equation:

$$\frac{dp_t}{dt} = -\mathrm{div}(f_t p_t) + \frac{g_t^2}{2} \Delta p_t \tag{39}$$

where $\Delta$ represents the Laplace operator (in $y$), namely $\mathrm{div}\nabla$, where $\nabla$ is the gradient operator (also in $y$). Rewriting this equation in the form of the continuity equation can be done as follows (Maoutsa et al., 2020a):

$$\frac{dp_t}{dt} = -\mathrm{div}\left(f_t p_t - \frac{g^2}{2}\frac{\nabla p_t}{p_t}p_t\right) = -\mathrm{div}\left(\left(f_t - \frac{g_t^2}{2}\nabla \log p_t\right)p_t\right) = -\mathrm{div}\left(w_t p_t\right)$$

where the vector field

$$w_t = f_t - \frac{g_t^2}{2}\nabla \log p_t \tag{40}$$

satisfies the continuity equation with the probability path $p_t$, and therefore generates $p_t$.

**Variance Exploding (VE) path**  The SDE for the VE path is

$$dy = \sqrt{\frac{d}{dt}\sigma_t^2}\,dw,$$

where $\sigma_0 = 0$ and increasing to infinity as $t \to 1$. The SDE is moving from data, $y_0$, at $t = 0$ to noise, $y_1$, at $t = 1$ with the probability path

$$p_t(y|y_0) = \mathcal{N}(y|y_0, \sigma_t^2 I).$$

The conditional VF according to equation 40 is:

$$w_t(y|y_0) = \frac{\sigma_t'}{\sigma_t}(y - y_0)$$

Using Lemma 1 we get that the probability path

$$\tilde{p}_t(y|y_0) = \mathcal{N}(y|y_0, \sigma_{1-t}^2 I)$$

is generated by

$$\tilde{w}_t(y|y_0) = -\frac{\sigma_{1-t}'}{\sigma_{1-t}}(y - y_0),$$

which coincides with equation 17.

**Variance Preserving (VP) path**  The SDE for the VP path is

$$dy = -\frac{T'(t)}{2}y + \sqrt{T'(t)}\,dw,$$

where $T(t) = \int_0^t \beta(s)ds$, $t \in [0,1]$. The SDE coefficients are therefore

$$f_s(y) = -\frac{T'(s)}{2}y, \quad g_s = \sqrt{T'(s)}$$

and

$$p_t(y|y_0) = \mathcal{N}(y|e^{-\frac{1}{2}T(t)}y_0, (1 - e^{-T(t)})I).$$

Plugging these choices in equation 40 we get the conditional VF

$$w_t(y|y_0) = \frac{T'(t)}{2}\left(\frac{y - e^{-\frac{1}{2}T(t)}y_0}{1 - e^{-T(t)}} - y\right) \tag{41}$$

Using Lemma 1 to reverse the time we get the conditional VF for the reverse probability path:

$$\tilde{w}_t(y|y_0) = -\frac{T'(1-t)}{2}\left(\frac{y - e^{-\frac{1}{2}T(1-t)}y_0}{1 - e^{-T(1-t)}} - y\right)$$

$$= -\frac{T'(1-t)}{2}\left[\frac{e^{-T(1-t)}y - e^{-\frac{1}{2}T(1-t)}y_0}{1 - e^{-T(1-t)}}\right],$$

which coincides with equation 19.

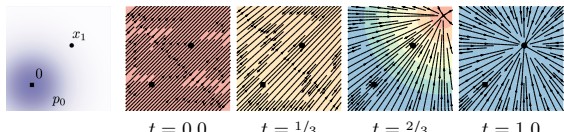

Diffusion path – conditional vector field

Figure 8: VP Diffusion path's conditional vector field. Compare to Figure 2.

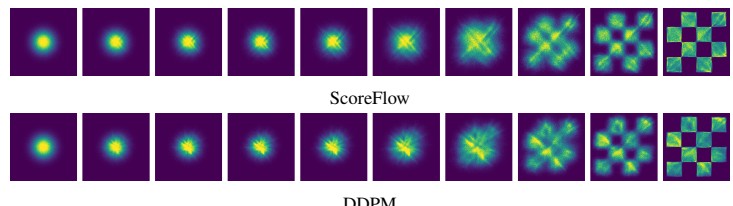

Figure 9: Trajectories of CNFs trained with ScoreFlow (Song et al., 2021) and DDPM (Ho et al., 2020) losses on 2D checkerboard data, using the same learning rate and other hyperparameters as Figure 4.

## E    IMPLEMENTATION DETAILS

For the 2D example we used an MLP with 5-layers of 512 neurons each, while for images we used the UNet architecture from Dhariwal & Nichol (2021). For images, we center crop images and resize to the appropriate dimension, whereas for the 32×32 and 64×64 resolutions we use the same pre-processing as (Chrabaszcz et al., 2017). The three methods (FM-OT, FM-Diffusion, and SM-Diffusion) are always trained on the same architecture, same hyper-parameters, and for the same number of epochs.

### E.1    DIFFUSION BASELINES

**Losses.**    We consider three options as diffusion baselines that correspond to the most popular diffusion loss parametrizations (Song & Ermon, 2019; Song et al., 2021; Ho et al., 2020; Kingma et al., 2021). We will assume general Gaussian path form of equation 10, *i.e.*,

$$p_t(x|x_1) = \mathcal{N}(x|\mu_t(x_1), \sigma_t^2(x_1)I).$$

Score Matching loss is

$$\mathcal{L}_{\text{SM}}(\theta) = \mathbb{E}_{t,q(x_1),p_t(x|x_1)}\lambda(t) \left\| s_t(x) - \nabla \log p_t(x|x_1) \right\|^2 \tag{42}$$

$$= \mathbb{E}_{t,q(x_1),p_t(x|x_1)}\lambda(t) \left\| s_t(x) - \frac{x - \mu_t(x_1)}{\sigma_t^2(x_1)} \right\|^2. \tag{43}$$

Taking $\lambda(t) = \sigma_t^2(x_1)$ corresponds to the original Score Matching (SM) loss from Song & Ermon (2019), while considering $\lambda(t) = \beta(1-t)$ ($\beta$ is defined below) corresponds to the Score Flow (SF) loss motivated by an NLL upper bound (Song et al., 2021); $s_t$ is the learnable score function. DDPM (Noise Matching) loss from Ho et al. (2020) (equation 14) is

$$\mathcal{L}_{\text{NM}}(\theta) = \mathbb{E}_{t,q(x_1),p_t(x|x_1)} \left\| \epsilon_t(x) - \frac{x - \mu_t(x_1)}{\sigma_t(x_1)} \right\|^2 \tag{44}$$

$$= \mathbb{E}_{t,q(x_1),p_0(x_0)} \left\| \epsilon_t(\sigma_t(x_1)x_0 + \mu_t(x_1)) - x_0 \right\|^2 \tag{45}$$

where $p_0(x) = \mathcal{N}(x|0, I)$ is the standard Gaussian, and $\epsilon_t$ is the learnable noise function.

**Diffusion path.**    For the diffusion path we use the standard VP diffusion (equation 19), namely,

$$\mu_t(x_1) = \alpha_{1-t}x_1, \quad \sigma_t(x_1) = \sqrt{1 - \alpha_{1-t}^2}, \quad \text{where } \alpha_t = e^{-\frac{1}{2}T(t)}, \quad T(t) = \int_0^t \beta(s)ds,$$

|  | CIFAR10 | ImageNet-32 | ImageNet-64 | ImageNet-128 |
|---|---|---|---|---|
| Channels | 256 | 256 | 192 | 256 |
| Depth | 2 | 3 | 3 | 3 |
| Channels multiple | 1,2,2,2 | 1,2,2,2 | 1,2,3,4 | 1,1,2,3,4 |
| Heads | 4 | 4 | 4 | 4 |
| Heads Channels | 64 | 64 | 64 | 64 |
| Attention resolution | 16 | 16,8 | 32,16,8 | 32,16,8 |
| Dropout | 0.0 | 0.0 | 0.0 | 0.0 |
| Effective Batch size | 256 | 1024 | 2048 | 1536 |
| GPUs | 2 | 4 | 16 | 32 |
| Epochs | 1000 | 200 | 250 | 571 |
| Iterations | 391k | 250k | 157k | 500k |
| Learning Rate | 5e-4 | 1e-4 | 1e-4 | 1e-4 |
| Learning Rate Scheduler | Polynomial Decay | Polynomial Decay | Constant | Polynomial Decay |
| Warmup Steps | 45k | 20k | - | 20k |

Table 3: Hyper-parameters used for training each model

with, as suggested in Song et al. (2020b), $\beta(s) = \beta_{\min} + s(\beta_{\max} - \beta_{\min})$ and consequently

$$T(s) = \int_0^s \beta(r)dr = s\beta_{\min} + \frac{1}{2}s^2(\beta_{\max} - \beta_{\min}),$$

where $\beta_{\min} = 0.1$, $\beta_{\max} = 20$ and time is sampled in $[0, 1-\epsilon]$, $\epsilon = 10^{-5}$ for training and likelihood and $\epsilon = 10^{-5}$ for sampling.

**Sampling.** Score matching samples are produced by solving the ODE (equation 1) with the vector field

$$u_t(x) = -\frac{T'(1-t)}{2}\left[s_t(x) - x\right]. \tag{46}$$

DDPM samples are computed with equation 46 after setting $s_t(x) = \epsilon_t(x)/\sigma_t$, where $\sigma_t = \sqrt{1 - \alpha_{1-t}^2}$.

### E.2    TRAINING & EVALUATION DETAILS

We report the hyper-parameters used in Table 3. We use full 32 bit-precision for training CIFAR10 and ImageNet-32 and 16-bit mixed precision for training ImageNet-64/128/256. All models are trained using the Adam optimizer with the following parameters: $\beta_1 = 0.9$, $\beta_2 = 0.999$, weight decay = 0.0, and $\epsilon = 1e-8$. All methods we trained (*i.e.*, FM-OT, FM-Diffusion, SM-Diffusion) using identical architectures, with the same parameters for the the same number of Epochs (see Table 3 for details). We use either a constant learning rate schedule or a polynomial decay schedule (see Table 3). The polynomial decay learning rate schedule includes a warm-up phase for a specified number of training steps. In the warm-up phase, the learning rate is linearly increased from $1e-8$ to the peak learning rate (specified in Table 3). Once the peak learning rate is achieved, it linearly decays the learning rate down to $1e-8$ until the final training step.

When reporting negative log-likelihood, we dequantize using the standard uniform dequantization. We report an importance-weighted estimate using

$$\log \frac{1}{K}\sum_{k=1}^{K} p_t(x + u_k), \text{ where } u_k \sim \mathcal{U}(0, 1), \tag{47}$$

with $x$ is in $\{0, \ldots, 255\}$ and solved at $t = 1$ with an adaptive step size solver `dopri5` with `atol=rtol=1e-5` using the `torchdiffeq` (Chen, 2018) library. Estimated values for different values of $K$ are in Table 4.

| Model | CIFAR-10 | | | ImageNet 32×32 | | | ImageNet 64×64 | | |
|---|---|---|---|---|---|---|---|---|---|
| | $K$=1 | $K$=20 | $K$=50 | $K$=1 | $K$=5 | $K$=15 | $K$=1 | $K$=5 | $K$=10 |
| *Ablation* | | | | | | | | | |
| DDPM | 3.24 | 3.14 | 3.12 | 3.62 | 3.57 | 3.54 | 3.36 | 3.33 | 3.32 |
| Score Matching | 3.28 | 3.18 | 3.16 | 3.65 | 3.59 | 3.57 | 3.43 | 3.41 | 3.40 |
| ScoreFlow | 3.21 | 3.11 | 3.09 | 3.63 | 3.57 | 3.55 | 3.39 | 3.37 | 3.36 |
| *Ours* | | | | | | | | | |
| FM $^{w}$/ Diffusion | 3.23 | 3.13 | 3.10 | 3.64 | 3.58 | 3.56 | 3.37 | 3.34 | 3.33 |
| FM $^{w}$/ OT | 3.11 | 3.01 | 2.99 | 3.62 | 3.56 | 3.53 | 3.35 | 3.33 | 3.31 |

Table 4: Negative log-likelihood (in bits per dimension) on the test set with different values of $K$ using uniform dequantization.

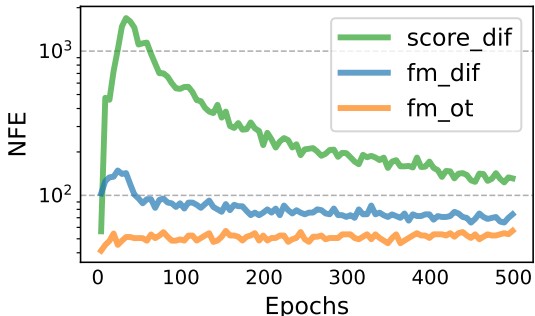

Figure 10: Function evaluations for sampling during training, for models trained on CIFAR-10 using `dopri5` solver with tolerance $1e^{-5}$.

When computing FID/Inception scores for CIFAR10, ImageNet-32/64 we use the TensorFlow GAN library [2]. To remain comparable to Dhariwal & Nichol (2021) for ImageNet-128 we use the evaluation script they include in their publicly available code repository [3].

# F    ADDITIONAL TABLES AND FIGURES

---

[2] https://github.com/tensorflow/gan
[3] https://github.com/openai/guided-diffusion

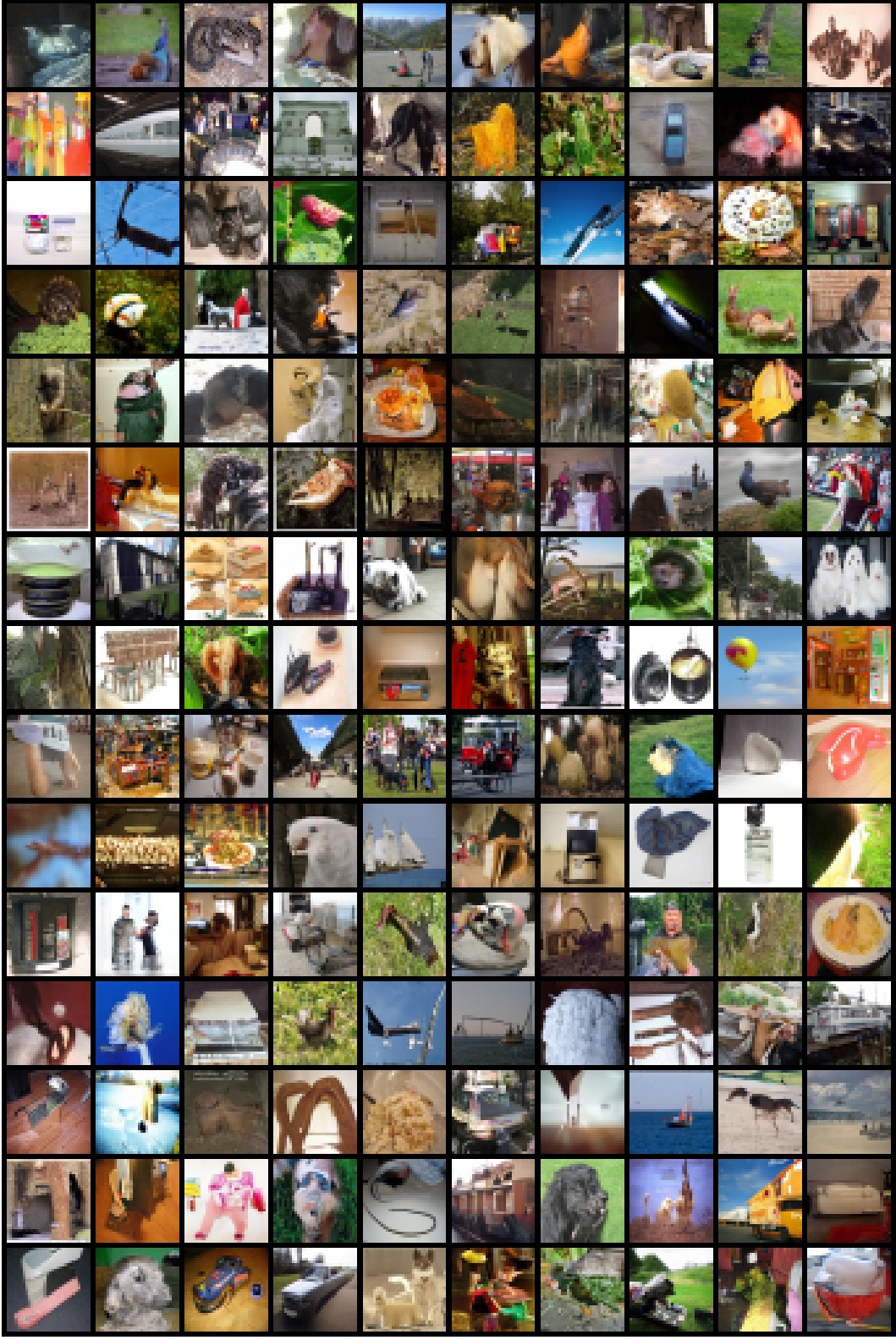

Figure 11: Non-curated unconditional ImageNet-32 generated images of a CNF trained with FM-OT.

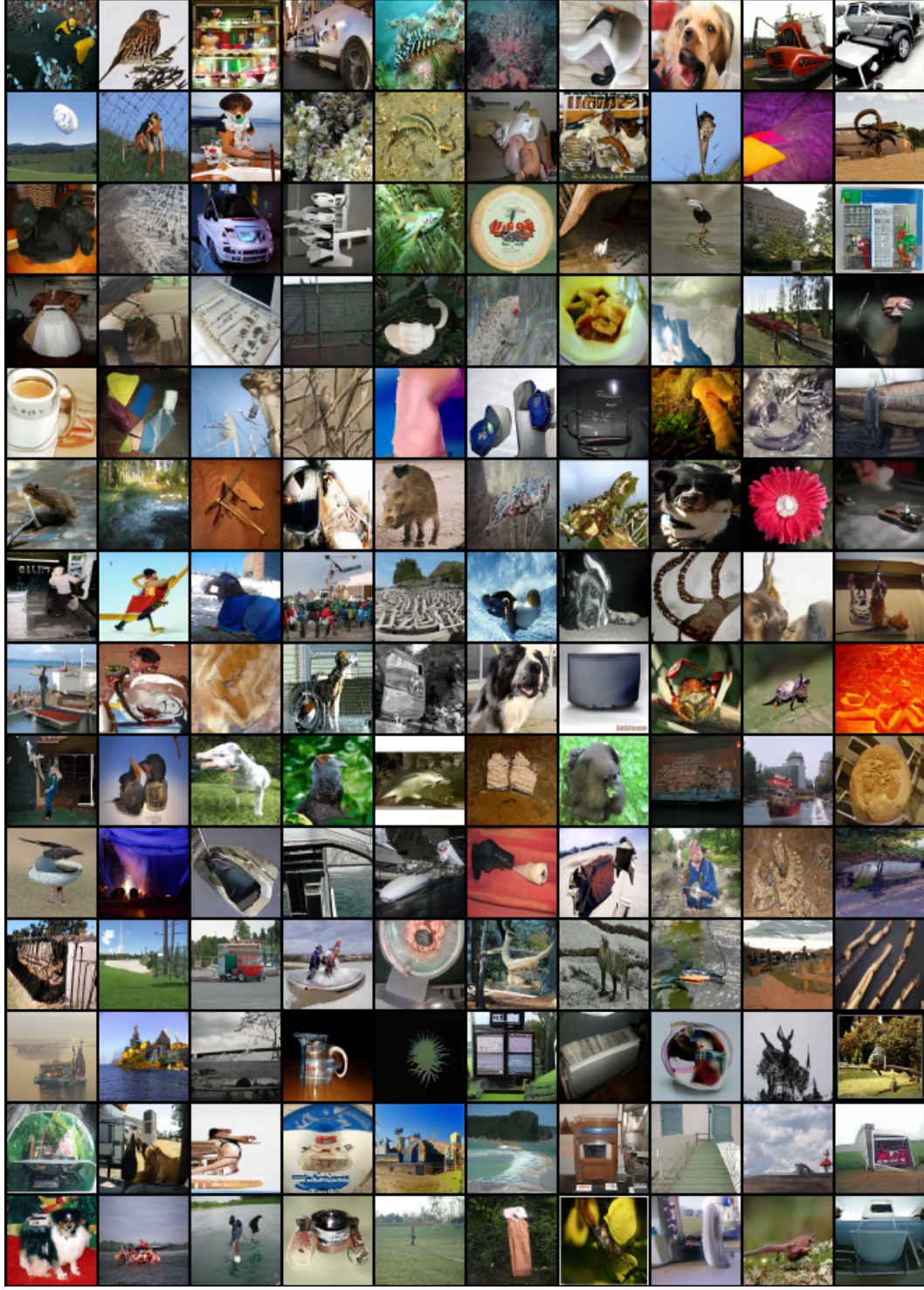

Figure 12: Non-curated unconditional ImageNet-64 generated images of a CNF trained with FM-OT.

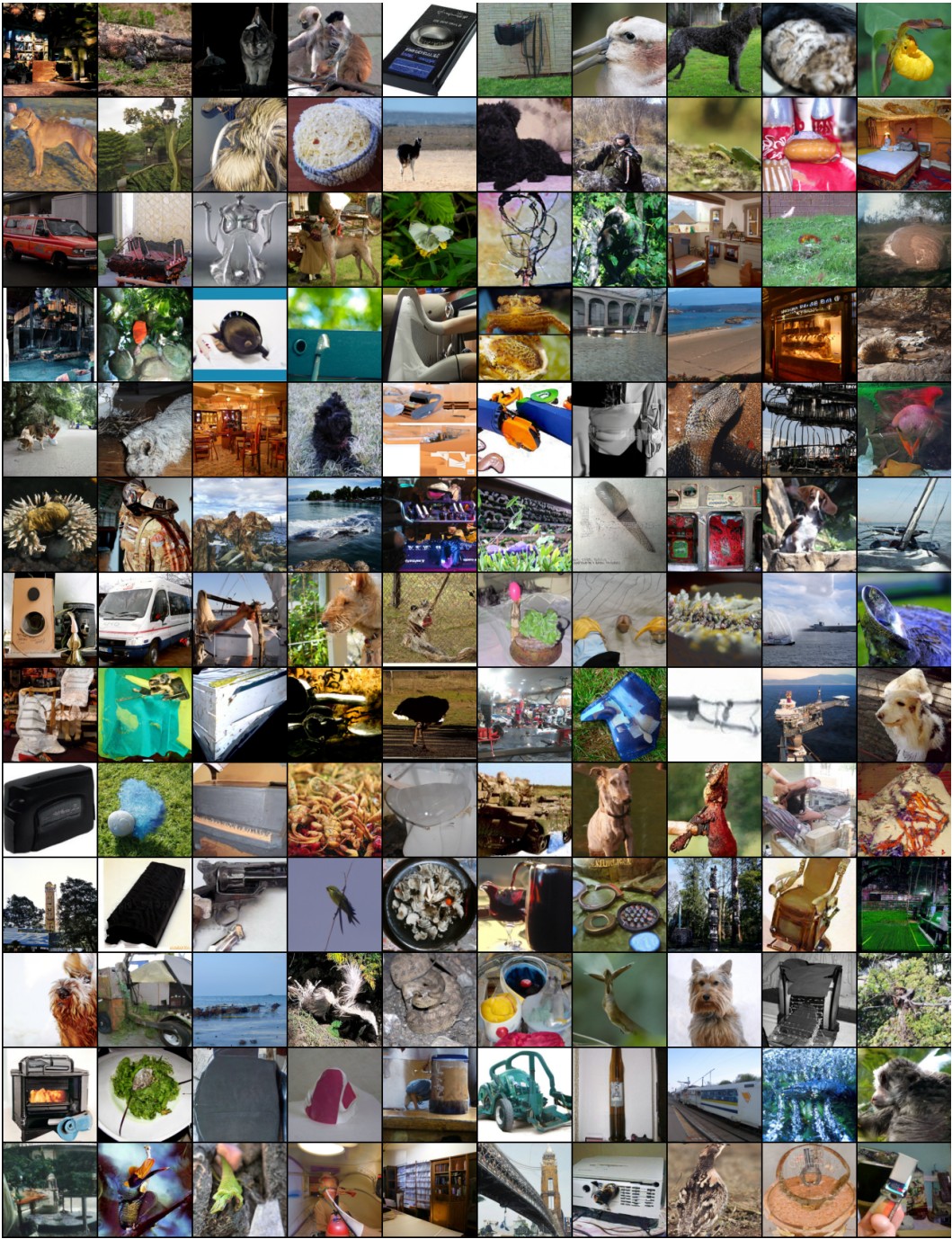

Figure 13: Non-curated unconditional ImageNet-128 generated images of a CNF trained with FM-OT.

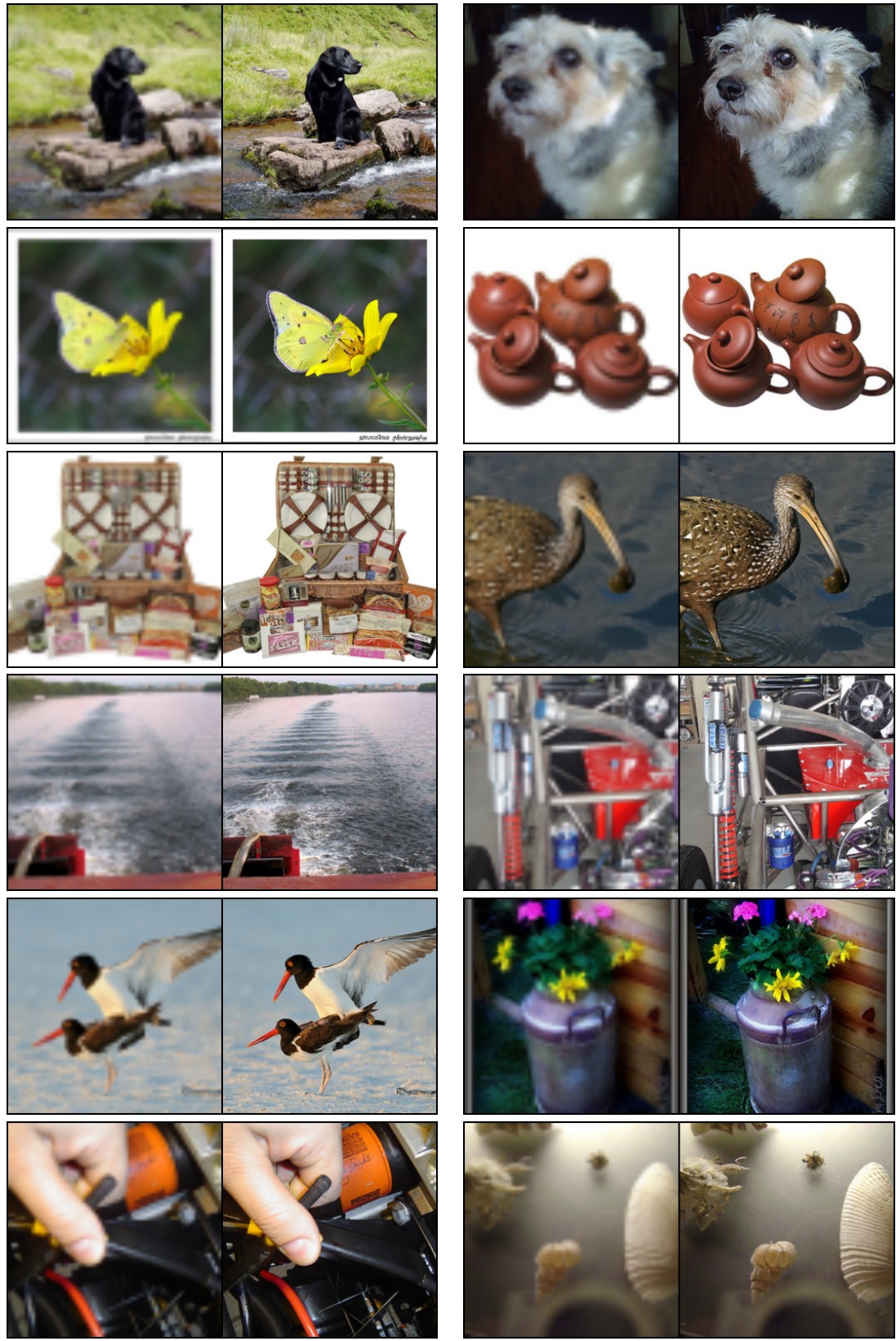

Figure 14: Conditional generation 64×64→256×256. Flow Matching OT upsampled images from validation set.

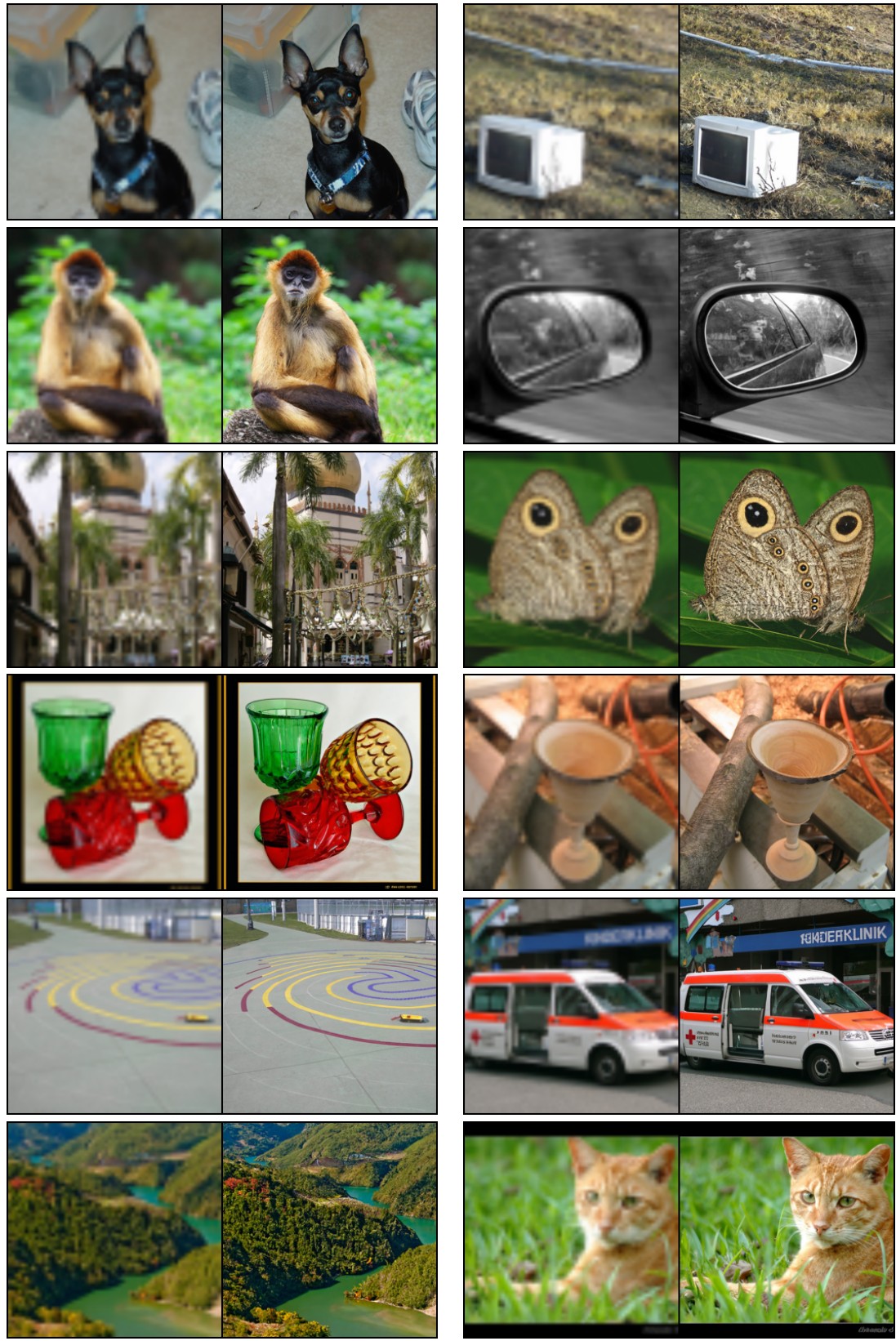

Figure 15: Conditional generation 64×64→256×256. Flow Matching OT upsampled images from validation set.

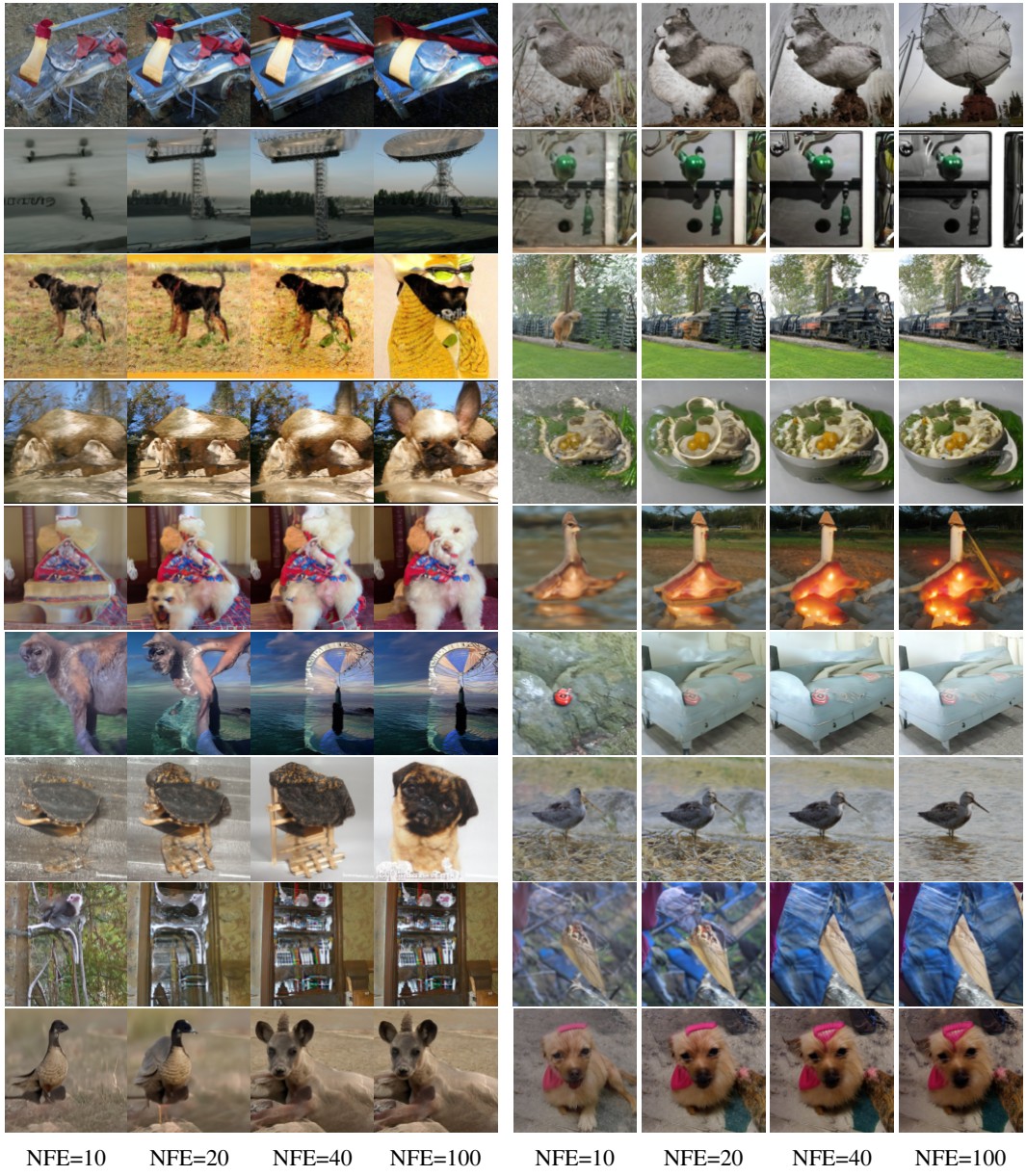

NFE=10          NFE=20          NFE=40          NFE=100          NFE=10          NFE=20          NFE=40          NFE=100

Figure 16: Generated samples from the same initial noise, but with varying number of function evaluations (NFE). Flow matching with OT path trained on ImageNet-128.

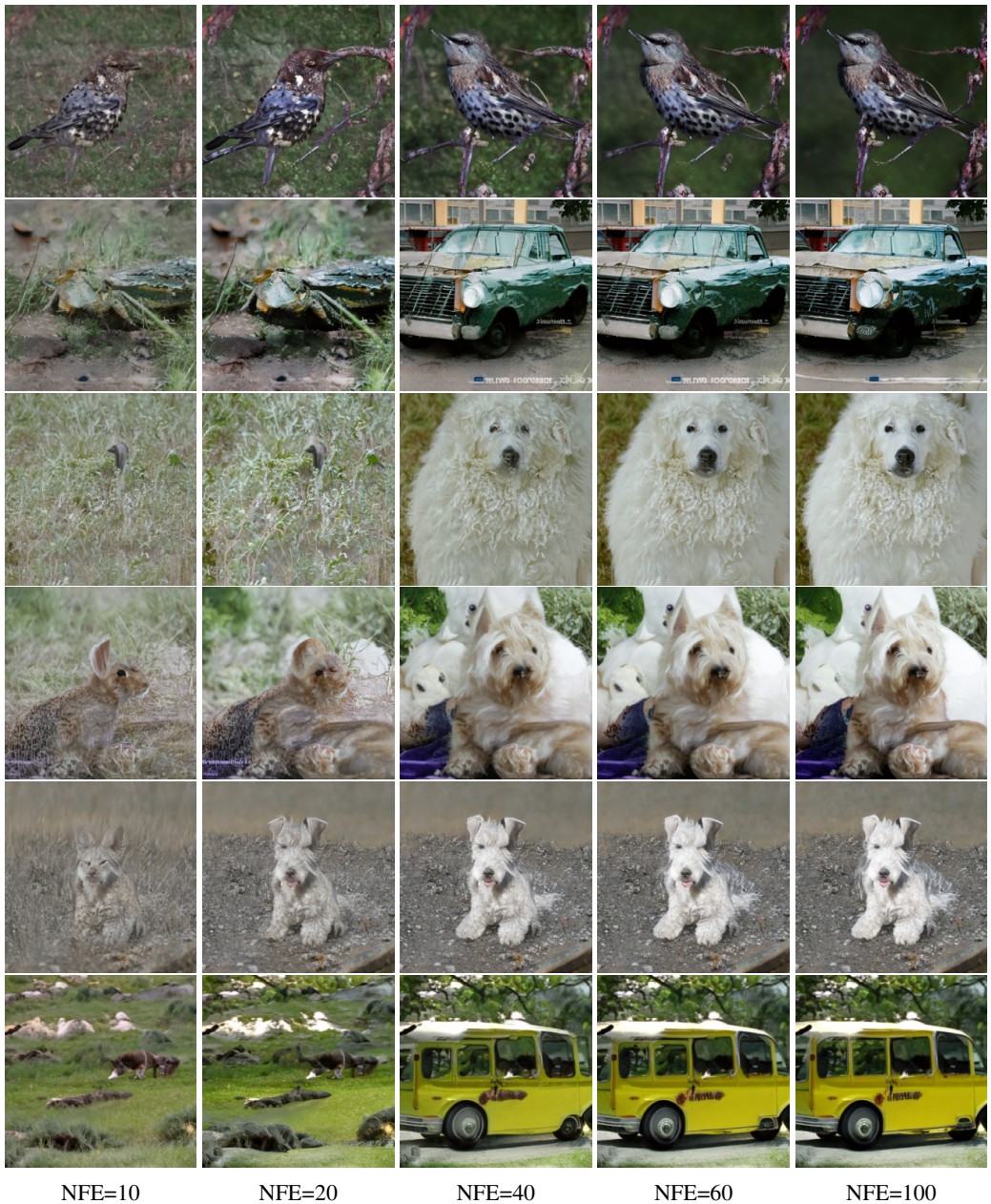

|  |  |  |  |  |
|---|---|---|---|---|
| NFE=10 | NFE=20 | NFE=40 | NFE=60 | NFE=100 |

Figure 17: Generated samples from the same initial noise, but with varying number of function evaluations (NFE). Flow matching with OT path trained on ImageNet 256×256.

