# OpenReview forum: "Flow Matching for Generative Modeling"
_ICLR.cc/2023/Conference — ICLR 2023 notable top 25%_

### Official Review · Reviewer_SQzg · 2022-10-23

**Confidence:** 4
**Correctness:** 4
**Technical Novelty And Significance:** 4
**Empirical Novelty And Significance:** 4
**Recommendation:** 10

**Clarity, Quality, Novelty And Reproducibility:**

- Clarity/quality: The paper was very high quality with respect to both theoretical and empirical results, and also clear with nice technical exposition and explanations.
- Novelty: The idea of matching vector fields is novel and interesting. The authors were also able to obtain impressive results.
- Reproducibility: No code was included with the submission.


**Strength And Weaknesses:**

Strengths:
- The empirical results are very strong: CNFs trained with OT-based FM achieves SOTA on likelihood estimation relative to existing baselines (though some of these competing methods do not actually compute likelihoods on that scale) and sample quality in terms of FID on ImageNet-{32,64}. They are competitive with existing baselines on CIFAR-10.
- OT-based FM also leads to more stable training, faster model convergence, and the simple/intuitive way in which the probability paths are defined leads to desirable properties such as “faster” (more efficient) generation of the desired sample starting from random noise. This is in contrast to diffusion models, where the perceptual quality of the denoised (generated) sample increases drastically towards the latter timesteps.
- Theoretically, the approach provides a nice perspective for understanding and unifying diffusion-based generative models. And by matching vector fields, this approach allows for the generalization of such generative models beyond the class of probability paths modeled by simple diffusions.


**Summary Of The Paper:**

This paper introduces Flow Matching (FM), which provides a way to scale Continuous Normalizing Flows (CNFs) to very high dimensions such as ImageNet-128. Rather than using expensive simulation-based training approaches (as for conventional CNF training), the authors propose to regress the vector field learned by a neural network against the true vector field given by predefined conditional probability paths. This approach for leveraging conditional probability paths, and also using optimal transport (OT) displacement maps for Gaussian conditional probability paths, not only leads to computational gains, but also provides an alternative perspective on diffusion-based generative models that improves their performance and stability as well.

**Summary Of The Review:**

Flow Matching is an interesting approach for scaling and improving the performance of not only CNFs, but also conventional diffusion-based generative models. The idea of matching vector fields and defining the generative process in terms of intuitive probability paths allows for not only empirical improvements in CNFs, but also ways to go beyond pre-defined diffusions and data modalities (e.g. this could be used for manifold data). This contribution is valuable and should be highlighted at the conference.

---

> ### Author Response · Authors · 2022-11-19
> **Response**
>
> Thank you for the review. We plan to release code upon publication for reproducibility and to ease adoption of our approach.

---

### Official Review · Reviewer_LJky · 2022-10-25

**Confidence:** 4
**Correctness:** 4
**Technical Novelty And Significance:** 3
**Empirical Novelty And Significance:** 3
**Recommendation:** 8

**Clarity, Quality, Novelty And Reproducibility:**

While [1] has already connected diffusion models to normalizing flows, this work further expands the class of learnable flows to include non-diffusion flows. In particular, it appears that OT flows are more efficient to sample from, which makes this extension more than a theoretical curiosity. Therefore, I believe the novelty is both significant and relevant.

Clarity:
The paper is generally well-written, though I found certain experimental results puzzling.

By my understanding, flow matching with diffusion flow (i.e. example I in Section 4.1) results in the same forward and reverse processes as diffusion models, except that, instead of just learning the score $\log p_t(\mathbf{x})$, flow matching models learn
$d\mathbf{x} = \left[ \mathbf{f}(\mathbf{x}, t) - \frac{1}{2} g(t)^2 \nabla_x \log p_t(\mathbf{x}) \right]dt$
i.e., Eq. 13 in [1].

However, according to Table 1, score matching and flow matching differ in performance, even when applied to the same diffusion process. Why is this the case? Shouldn't the learned ODE be identical between flow and score matching? In a similar vein, why would Flow Matching perform so much worse than score matching on CIFAR-10 in general?

**Strength And Weaknesses:**

Strengths:
- The work deepens the connections between diffusion models and normalizing flows originally proposed in [1], and allows for more types of probability flows, including non-diffusion flows.
- Training involves a simple drop-in replacement of the loss in score-matching diffusion models.
- Certain choices of the learned flow greatly improve sampling speed during inference.

Weaknesses:
- Models do not exhibit improved performance on CIFAR-10.
- Authors claim state-of-the-art performance on ImageNet in terms of sample quality, however, FID metrics are much higher than that reported in [2] (see Table 5, for ImageNet 64x64). Can the authors comment on this?

[1] Song, Y., Sohl-Dickstein, J., Kingma, D.P., Kumar, A., Ermon, S. and Poole, B., 2020. Score-based generative modeling through stochastic differential equations. arXiv preprint arXiv:2011.13456.

[2] Dhariwal, P. and Nichol, A., 2021. Diffusion models beat gans on image synthesis. Advances in Neural Information Processing Systems, 34, pp.8780-8794.

**Summary Of The Paper:**

This paper introduces a new framework for training continuous normalizing flows, inspired by recent work on diffusion models. Originally, normalizing flows are trained by maximizing the likelihood of the pushforward density, which requires simulating the flow end-to-end for each gradient update and can thus be very costly. In this work, the proposed "flow matching" framework involves approximating the time-varying vector field of a desired ODE for each time t, much like the score matching algorithm in diffusion models. Thus training no longer requires end-to-end simulation of the flow, and simply involves a squared error loss for each time $t$.


**Summary Of The Review:**

The paper is well-written and provides a straightforward generalization of the diffusion framework. Performance on ImageNet 64x64 is strong, and suggests that the theory translates well into practical improvements in image modeling.

---

> ### Author Response · Authors · 2022-11-19
> **Response**
>
> Thank you for the review. Below we address the questions and comments raised in the review.
>
> **Q:** Authors claim state-of-the-art performance on ImageNet in terms of sample quality, however, FID metrics are much higher than that reported in [2] (see Table 5, for ImageNet 64x64). Can the authors comment on this?
>
> **A:** Our state-of-the-art results on ImageNet are for the unconditional image synthesis setting. In contrast, the results for ImageNet 64x64 reported in [2] are for a class conditioned model, using additional information by incorporating class labels into the model (e.g., via normalization layers), which makes the generation task easier and incomparable to the unconditional setting.
>
>
> **Q:** By my understanding, flow matching with diffusion flow (i.e. example I in Section 4.1) results in the same forward and reverse processes as diffusion models, except that, instead of just learning the score log⁡pt(x), flow matching models learn dx=[f(x,t)−1/2 g(t)2∇xlog⁡pt(x)]dt
>  i.e., Eq. 13 in [1].
> However, according to Table 1, score matching and flow matching differ in performance, even when applied to the same diffusion process. Why is this the case? Shouldn't the learned ODE be identical between flow and score matching?
>
> **A:** It is true that the Flow Matching with diffusion flow (FM-Dif) aims to regress the same Diffusion process as previous diffusion models, however it formulates the optimization problem using a different loss. The solutions to these different loss parameterizations coincide only for Loss=0, which is never the case in practice. In fact, many works on diffusion models differ in the way they parametrize the loss for the same diffusion process and report this to have significant practical implications. FM-Dif can be seen as another parametrization that directly regresses the probability path’s generating vector field.
>
> **Q:**  Why would Flow Matching perform so much worse than score matching on CIFAR-10 in general?
>
> **A:** Our experiments (e.g., Table 1 (left) in the revised paper) show Flow Matching consistently improves performance over existing Diffusion methods when compared on the same model architecture. Furthermore, our results on CIFAR-10 are comparable to the results of Score-Matching. For Score-Flow we were not able to reproduce their best Baseline results (2.95 BPD). Lastly, we note that we have used a model architecture tuned on ImageNet [2], which potentially overfits on CIFAR-10.
>
> [2] Dhariwal, P. and Nichol, A., 2021. Diffusion models beat gans on image synthesis. Advances in Neural Information Processing Systems, 34, pp.8780-8794.

---

### Official Review · Reviewer_tWC7 · 2022-10-30

**Confidence:** 4
**Correctness:** 3
**Technical Novelty And Significance:** 4
**Empirical Novelty And Significance:** 3
**Recommendation:** 8

**Clarity, Quality, Novelty And Reproducibility:**

**Clarity**
The paper is clearly written and presented. The diagrams throughout are informative and helpful.

**Novelty**
As far as I'm aware, the core method for directly and efficiently regressing the vector field corresponding to a probability path is novel, as are the connections to diffusion models and optimal transport.

**Strength And Weaknesses:**

**Strengths**
The fundamental idea proposed in the paper is excellent. Analogously to how denoising score-matching makes score-matching tractable in diffusion models, flow matching shows that the diffusion formulation is not needed at all, and regressing the target vector field defined by a simple probability path is possible directly. The authors then explicitly lay out the connection to existing diffusion models, as well as an alternative transport path. Both the flow matching objective and the OT path are ablated independently on toy and ImageNet-level tasks. These are all strong contributions, potentially simplifying the generative modeling landscape and making strong connections in the existing literature (continuous-time flows, diffusion models, optimal transport).

**Weaknesses**
A small constant sigma_min > 0 is used throughout the paper as the initial noise applied to the data distribution. That is, t = 1 corresponds to sigma =  sigma_min instead of sigma = 0. As far as I can see, the reason for introducing sigma_min is not stated explicitly. Is it there for practical/numerical or theoretical reasons? Why can't we have sigma = 0 at t = 1? This should be explained explicitly.

I think the biggest difficulty I have with the paper is the baseline diffusion model used for the empirical evaluation. A key claim of the paper is that the proposed method effectively gives practitioners a better way to fit the vector field of a probability path than through the diffusion formulation. I would therefore expect the implementation of the diffusion baseline to be treated with due care.

For example, appendix E.1 eq (41) states that the diffusion model regresses the score function directly. In practice, diffusion models often target a quantity related to the score, such as the clean data, or the Gaussian noise added to the clean data, from which the score is derived. The choice of parameterization can significantly impact the quality of the learned model (DDPM, Ho et al. 2020), since the magnitude of the score varies widely over time. Moreover, the ODE in eq. 42 has semi-linear structure (the sum of a linear and non-linear term). Existing work (DDIM, Song et al. 2020, DPM-solver, Lu et al. 2022) has shown that using black-box solvers which don't account for this structure (i.e. solving the linear part exactly) can lead to unnecessary errors in the solution, again impacting performance.

If it is an advantage of flow matching that the parameterization of the regression target and the choice of ODE solver do not need such careful consideration, this should be stated explicitly and taken into account for direct comparison, rather than comparing to a diffusion model implementation which is not as performant as it might be.

Nits:
- 4: 'There is potentially an infinite number of vector fields that generate any particular probability path,
but the vast majority of these' -- what is meant by vast majority? Quantify this formally.
- 4.1: 'we can set them to any reasonable function' -- what is a 'reasonable function'? Again be precise.
- 6.1: 'we find that we can achieve very reasonable performance' -- what is 'very reasonable performance'?
- 6.1: The BigGAN citation is for Lučić et al. 2019 but the canonical citation is Brock et al. 2018
- 7: 'FM can be generalized to manifold data' -- perhaps, but you can't claim this without demonstrating it

**Summary Of The Paper:**

The paper presents flow matching, a method for training continuous-time normalizing flows by directly regressing the vector field of a chosen probability path. Flow matching only requires the ability to sample from the chosen probability path for training, and importantly does not require propagating gradients through ODE solver dynamics. The authors connect flow matching to optimal transport and existing diffusion methods for generative modeling. Empirical results suggest flow matching can give improved training and sampling efficiency over standard diffusion models.

**Summary Of The Review:**

While there are a couple of issues I would like to see addressed, I think this is a strong submission and would recommend acceptance.

---

> ### Author Response · Authors · 2022-11-19
> **Response**
>
> Thank you for the review. Below we address the questions and comments raised in the review.
>
> **Q:** I think the biggest difficulty I have with the paper is the baseline diffusion model used for the empirical evaluation… In practice, diffusion models often target a quantity related to the score, such as the clean data, or the Gaussian noise added to the clean data, from which the score is derived. The choice of parameterization can significantly impact the quality of the learned model (DDPM, Ho et al. 2020), since the magnitude of the score varies widely over time.
>
> **A:** Thank you for this comment. We have now expanded our baseline set of experiments to include other parametrizations of the diffusion loss. In particular, we added 2 more variations: the max-likelihood score matching [1], and the DDPM (noise matching) loss formulation of [2]. Together with the existing score-matching loss [3] we feel these cover most popular formulations. Please see Table 1 (left), Figure 5, and expanded implementation details in Appendix E.1 in the revised paper for details.
>
>
>
> **Q:** Moreover, the ODE in eq. 42 has semi-linear structure (the sum of a linear and non-linear term). Existing work (DDIM, Song et al. 2020, DPM-solver, Lu et al. 2022) has shown that using black-box solvers which don't account for this structure (i.e. solving the linear part exactly) can lead to unnecessary errors in the solution, again impacting performance.
>
> **A:** As noted in [4] (see Figure 2a, for example) using higher order solver and/or sufficiently small step size (large NFE) results in a comparable or even better sample quality compared to specialized fast solvers. The main benefit of these specialized solvers is their ability to provide good sampling with low NFEs. In our ablations we sampled all methods with the dopri5 method with tolerance eps=10-5, which compares RK4 and RK5 solutions to control error. We also verified on small datasets that further decreasing the tolerance does not significantly change the ode solution for BPD computation. We therefore believe our solver is *as performant as the best solver out there* in terms of accuracy.
>
>
>
>
> [1] Yang Song, Conor Durkan, Iain Murray, and Stefano Ermon. Maximum likelihood training of score-based diffusion models. In Thirty-Fifth Conference on Neural Information Processing Systems, 2021.
>
> [2] Jonathan Ho, Ajay Jain, and Pieter Abbeel. Denoising diffusion probabilistic models.
> Advances in Neural Information Processing Systems, 33:6840–6851, 2020.
>
> [3] Yang Song, Jascha Sohl-Dickstein, Diederik P Kingma, Abhishek Kumar, Stefano Ermon, and Ben Poole. Score-based generative modeling through stochastic differential equations.
> arXiv preprint arXiv:2011.13456, 2020b.
>
> [4] DPM-Solver: A Fast ODE Solver for Diffusion Probabilistic Model Sampling in Around 10 Steps

---

### Official Review · Reviewer_AKwV · 2022-11-04

**Confidence:** 4
**Correctness:** 2
**Technical Novelty And Significance:** 2
**Empirical Novelty And Significance:** 3
**Recommendation:** 5

**Clarity, Quality, Novelty And Reproducibility:**

* Very clear writing.

* Quality in experiments and overall novelty is less satisfying. See detailed explanation above.

**Strength And Weaknesses:**

## Strength
1. Very clear writing.
2. Proposed method is simple and effective.
3. The method generalizes conventional score-based diffusion models without complicating the formulation.
4. The optimal transport path stabilizes training and leads to more efficient sampling.

## Weaknesses

1. Novelty is limited. One central contribution of this work is a method to aggregate conditional probability paths to form a target probability path that connects a simple distribution $p_0$ to the data distribution $p_1$. This is actually a special case of the result in [1]. In fact, Theorem 1 in this paper can be derived from Theorem 1 in [1] by simply replacing SDEs with ODEs (setting the diffusion coefficient to zero). In addition, the conditional flow matching objective was already proposed in [1]. In light of [1], the contributions of this work are mostly experimental verifications, and using optimal transport to construct probability paths. It is disappointing to see that this work does not cite [1] and neither does it discuss the important connections.

2. Experiments on ImageNet are likely wrong. It is surprising to see that Flow Matching has a significant lead over other approaches on ImageNet 32x32 and 64x64, while having mediocre results on CIFAR-10. Even the authors' reproduction of score matching models on ImageNet is significantly better than previously reported. I'm 99% sure that this is caused by using the wrong downsampled ImageNet dataset. In generative modeling, people use the downsampled ImageNet dataset from [2], not the one from [3]. The major difference is in downsampling algorithms: the former does not use anti-aliasing and is therefore significantly more difficult for maximum likelihood training compared to the latter. Authors need to re-run their experiments on ImageNet and report the correct results.

3. Authors motivate Flow Matching as a simulation-free training method for continuous normalizing flows. It is important to mention that ScoreFlow in [4] was actually proposed earlier with the same goal. In fact, ScoreFlow is the first simulation-free training method for continuous normalizing flows that are based on diffusion processes. Compared to flow matching, it has the advantage of directly maximizing log-likelihood (see also [5]) and thus will outperform flow matching in terms of NLLs. This paper should compare with ScoreFlow more carefully, include more discussions with it, and highlight aspects where flow matching outperforms ScoreFlow.

## References
[1] Peluchetti, Stefano. "Non-Denoising Forward-Time Diffusions." (2021).

[2] Van Den Oord, Aäron, Nal Kalchbrenner, and Koray Kavukcuoglu. "Pixel recurrent neural networks." International conference on machine learning. PMLR, 2016.

[3] Chrabaszcz, Patryk, Ilya Loshchilov, and Frank Hutter. "A downsampled variant of imagenet as an alternative to the cifar datasets." arXiv preprint arXiv:1707.08819 (2017).

[4] Song, Yang, et al. "Maximum likelihood training of score-based diffusion models." Advances in Neural Information Processing Systems 34 (2021): 1415-1428.

[5] Lu, Cheng, et al. "Maximum likelihood training for score-based diffusion odes by high order denoising score matching." International Conference on Machine Learning. PMLR, 2022.

**Summary Of The Paper:**

This work proposes flow matching, a method for training continuous normalizing flows without numerical ODE simulation. Key to flow matching is a framework for constructing suitable probability paths via a mixture of conditional (simpler) probability paths. Authors further draw connection to score-based diffusion models and optimal transport. Inspired by the connection with optimal transport, authors propose a new type of probability paths that are more efficient than existing ones in diffusion models. Experiments confirm that the proposed flow matching approach can obtain competitive image quality and likelihoods on real-world image datasets.

**Summary Of The Review:**

Flow matching is a nice and clean formulation for simulation-free training of CNFs, generalizing score-based diffusion models to non-Gaussian noise and optimal transport paths. However, the idea is not completely new, and experimental results on ImageNet are unreliable.

---

> ### Author Response · Authors · 2022-11-19
> **Response**
>
> Thank you for the review. Below we address the questions and comments raised in the review.
>
> **Q:** Novelty is limited. Theorem 1 in this paper can be derived from Theorem 1 in [1] by simply replacing SDEs with ODEs (setting the diffusion coefficient to zero). In addition, the conditional flow matching objective was already proposed in [1]. In light of [1], the contributions of this work are mostly experimental verifications, and using optimal transport to construct probability paths. It is disappointing to see that this work does not cite [1] and neither does it discuss the important connections.
>
> **A:** We respectfully disagree: we think Flow Matching is neither a special case nor a simple instance of [1]. We do agree, however, a citation to [1] would be helpful and added a reference near Theorem 1 and in the Related works section.
> In more detail, we would like to clarify our key contributions over [1]:
>
> - **Flow Matching works with a wider class of probability density paths** - while [1] constructs paths using diffusion bridges which are in particular diffusion processes, Flow Matching provides a framework for designing a wider class of probability density paths. Indeed every diffusion process’s probability law in particular satisfies the continuity equation with a suitable vector field (see e.g., equations 37, 38, and 39 in Appendix). The optimal transport path proposed in our paper goes beyond Diffusion processes used in previous works.
>
> - **Flow Matching objective does not appear in [1]** - The suggested objectives in [1] are either score matching (equation 24) or regressing the conditional expectations of clean images (equations 25, 26); neither are equivalent to regressing the conditional generating vector field.
>
> - **Simplified theory based on ODEs** - Flow Matching is derived from first principles of dynamics driven by an ODE. We believe that in addition to being more general (not limited to diffusion paths), viewing diffusion models through the lens of Flow Matching serves the community by providing a simpler and accessible entry point for future research.
> Regarding Theorem 1 in [1], it is indeed related to our work,  i.e., [1]  provides a marginal drift and diffusion formulation for a mixture of diffusion processes similar to our construction of a mixture of conditional probability paths, and we highlighted that in our revised paper (see after Theorem 1).
>
>
> **Q:** Experiments on ImageNet are likely wrong. It is surprising to see that Flow Matching has a significant lead over other approaches on ImageNet 32x32 and 64x64, while having mediocre results on CIFAR-10. Even the authors' reproduction of score matching models on ImageNet is significantly better than previously reported. I'm 99% sure that this is caused by using the wrong downsampled ImageNet dataset. In generative modeling, people use the downsampled ImageNet dataset from [2], not the one from [3]. The major difference is in downsampling algorithms: the former does not use anti-aliasing and is therefore significantly more difficult for maximum likelihood training compared to the latter. Authors need to re-run their experiments on ImageNet and report the correct results.
>
> **A:** Thank you for pointing out this issue, we were not aware of a second ImageNet 32/64 dataset. Unfortunately, the dataset of [2] (OORD et al. 2016) is no longer available for download in the ImageNet repository. We worked with the only downsampled ImageNet-32/64 that is available for download from the ImageNet repository. We do realize some or all of the previous works have benchmarked the old datasets. We therefore recreated the main baselines and ran them on the dataset of [3] (Chrabaszcz et al. 2017) achieving two goals: (1) establishing baseline results on this downsample ImageNet (currently the only available) datasets; and (2) comparing the different methods on the *same* architecture, and therefore focusing on our paper’s contributions which is the training approach rather than the architecture. Please note that the preprocessing for our ImageNet 128 experiment is comparable to previous works and provides SOTA performance, which we have now highlighted more strongly in the paper.
>
> **Q:** Compared to flow matching, [ScoreFlow] has the advantage of directly maximizing log-likelihood (see also [5]) and thus will outperform flow matching in terms of NLLs. This paper should compare with ScoreFlow more carefully, include more discussions with it, and highlight aspects where flow matching outperforms ScoreFlow.
>
> **A:** ScoreFlow minimizes merely an upper bound to the NLL and therefore is not guaranteed to produce better likelihoods. We have incorporated a comparison to ScoreFlow with the Maximum Likelihood weighting in our baselines, please see Table 1 (left) and Figure 5.

---

### Author Response · Authors · 2022-11-19
**Summary of changes in revision**

We thank the reviewers for their detailed reviews. We have addressed the individual questions, comments and concerns in separate threads. Here we summarize the main changes in the revised paper for the convenience of all reviewers and AC.

**1)** Added a conditional image generation experiment: image super-resolution 64x64->256x256 showing considerably improved FID and IS results over the state-of-the-art model. See Section 6.3, Table 2 and Figures 14 and 15.

**2)** Table 1 (left): Added 2 more Diffusion baselines (DDPM [1] and Maximum-Likelihood ScoreFlow [2]) completing a set of 3 of the most popular Diffusion baselines (including [3]), all tested on the same model architecture. Removed previous baselines that are potentially using a no longer available version of the downsampled ImageNet dataset (see more details in the answer to Reviewer AKwV).

**3)** Added Table 1 (right) for ImageNet 128x128 showing state-of-the-art unconditional generation results on this dataset compared to relevant baselines.

**4)** Added a relevant reference to [4] after Theorem 1, and Diffusion Bridges methods in the related work section (Section 5) .

**5)** Added Figure 5 showing image quality (FID) progression during training of all methods.

As a final comment: we plan to make our code publicly available to aid reproducibility, especially with regards to the imagenet experiments.

—--------------------------------------------------------------------------------------------------------------

[1] Yang Song, Conor Durkan, Iain Murray, and Stefano Ermon. Maximum likelihood training of score-based diffusion models. In Thirty-Fifth Conference on Neural Information Processing Systems, 2021.

[2] Jonathan Ho, Ajay Jain, and Pieter Abbeel. Denoising diffusion probabilistic models.
Advances in Neural Information Processing Systems, 33:6840–6851, 2020.

[3] Yang Song, Jascha Sohl-Dickstein, Diederik P Kingma, Abhishek Kumar, Stefano Ermon, and Ben Poole. Score-based generative modeling through stochastic differential equations.
arXiv preprint arXiv:2011.13456, 2020b.

[4] Peluchetti, Stefano. "Non-Denoising Forward-Time Diffusions." (2021).

---

### Decision · Program_Chairs · 2023-01-20

**Decision:**

Accept: notable-top-25%

**Justification For Why Not Higher Score:**

Not oral because the empirical results are not uniformly stellar in terms of quality, though speed/stability gains seem impressive.

**Justification For Why Not Lower Score:**

Spotlight because diffusion models are are of intense interest currently, and this paper proposes an alternative, simpler and arguably more effective scheme with strongly positive reviews.

**Metareview: Summary, Strengths And Weaknesses:**

Summary: This paper proposes a new approach for training continuous-time normalizing flows by directly estimating a vector field that generates a simple pre-defined target probability path. The paper argues that this approach is more general and efficient than diffusion.

Strengths: Simple and effective method that is surprisingly competitive. Strong empirical results particularly in terms of more stable training, faster model convergence and sampling.

Weaknesses:  Models do not exhibit substantially improved performance on CIFAR-10. Baseline diffusion models could perhaps be strengthened.

**Note From Pc:**

if the above contains the word "oral" or "spotlight" please see: "oral" presentation means -> notable-top-5% and "spotlight" means -> notable-top-25%. As stated in our emails, we are disassociating presentation type from AC recommendations